# Assessment of surface ozone products from downscaled CAMS reanalysis and CAMS daily forecast using urban air quality monitoring stations in Iran

Najmeh Kaffashzadeh[1] and Abbas Ali Aliakbari Bidokhti[1]

[1] Institute of Geophysics, University of Tehran, Tehran, Iran

*Correspondence to*: Kaffashzadeh Najmeh (najmeh.kaffashzadeh@gmail.com)

**Abstract.** Tropospheric ozone time series consist of the effects of various scales of motion, from meso to large timescales, which is often challenging for global models to capture. This study uses two global datasets, namely the reanalysis and daily forecast of the Copernicus Atmosphere Monitoring Service (CAMS), to assess the capability of these products in presenting ozone's features on regional scales. We obtained 16 relevant meteorological and several pollutant species, such as $O_3$, CO, $NO_x$, etc., from CAMS. Furthermore, we employed a comprehensive set of in situ measurements of ozone at 27 urban stations in Iran for the year 2020. We decomposed the time series into three spectral components, i.e., short (S), medium (M), and long (L) terms. To cope with the scaling issue between the measured data and the CAMS' products, we developed a downscaling approach based on a Long Short-Term Memory (LSTM) neural network method, with, apart from modeled ozone, also assimilated meteorological quantities, as well as lagged $O_3$ observations. Results show the benefit of applying the LSTM method instead of using the original CAMS products for providing $O_3$ over Iran. It is found that lagged $O_3$ observation has a larger contribution than other predictors in improving the LSTM. Comparing to the S, the M component shows more associations with observations, e.g., correlation coefficients larger than 0.7 for S and about 0.95 for M in both models. The performance of the models varies across cities; for example, the highest error is for areas with high emissions of $O_3$ precursors. The robustness of the results is confirmed by performing an additional downscaling method. This study demonstrates that coarse-scale global model data such as CAMS needs to be downscaled for regulatory purposes or policy applications at local scales. Our method can be useful not only for the evaluation but also for the prediction of other chemical species, such as aerosols.

## 1 Introduction

Near-surface ozone ($O_3$) is a secondary air pollutant that deteriorates human health and plants via damaging respiratory systems (Bell et al., 2006; Fowler et al., 2009; Mills et al., 2011; Malley et al., 2015; Pozzer et al., 2023). Exposure to high concentrations of air pollution, especially $O_3$, leads to premature deaths, in particular for those suffering from asthma disease. Many efforts have been made to study ozone and its precursors in Iran, which suffers from severe ambient air

pollution (Lelieveld et al., 2009; Bidokhti et al., 2016; Faridi et al., 2018; Yousefian et al., 2020). As an example, Hadei et al.

(2017) reported a total of 1363 premature deaths attributed to $O_3$ in Tehran within three years, 2013–2016. Long-term exposures to ambient $O_3$ are responsible for 173 deaths from respiratory disease in Ahvaz for the year 2012 (Goudarzi et al., 2015).

Ozone is either transported naturally from the stratosphere or produced in situ by photochemical oxidation of ozone's precursor gases such as nitrogen oxides ($NO_x$), non-methane volatile organic compounds (NMVOC), methane ($CH_4$), or carbon

monoxide (CO) in the presence of sunlight (Crutzen 1974; Monks et al., 2015; Cooper et al., 2014). The ozone levels are not only a function of its precursor's emissions but also of meteorological conditions that influence the evolution of emissions, depositions, and photochemical products (Bloomer et al., 2009; Li et al., 2020). It has been shown that not only local emissions and winds but also synoptic conditions control the ozone levels over Iran (Borhani et al., 2021; Zohdirad et al., 2022; Jafari Hombari and Pazhoh, 2022). Several synoptic systems, which cause the high levels of ozone over Tehran, have been

recognized and classified in a study by Khansalari et al. (2020) and Lashkari et al. (2020).

Reanalysis data provide a global picture of past weather and climate. These data are constructed by combining atmospheric observations such as satellites, radar, and in situ measurements with a detailed simulation of the atmosphere, using data assimilation techniques. Reanalysis data have been widely used as an initial condition for the daily forecast of the atmosphere or boundary conditions in regional models, for the study of climate change, and as proxies to complement insufficient in situ

measurements. In recent years, the Copernicus Atmosphere Monitoring Service (CAMS) has been mainly developed to assimilate observations of chemical compositions to provide analyses of tropospheric ozone and aerosol concentrations, but it also holds outputs for several meteorological variables (Innes et al., 2019). Several studies have evaluated CAMS reanalysis (hereafter CAMSRA) products and compared them with other reanalysis datasets and a control run (without assimilation of atmospheric composition). As an example, an intercomparison of tropospheric ozone from seven reanalysis datasets in East

Asia has reported that CAMSRA depicts more reasonable spatial-temporal variability than other datasets (Park et al., 2020). They also show the suitability of CAMSRA for the study of local tropospheric ozone on seasonal to interannual timescales but the inadequacy of that to study long-term trends. Results of the study by Huijnen et al. (2020) reveal the ability of CAMSRA to reproduce background $O_3$ in terms of mean and variability on various timescales such as synoptic, seasonal, etc. Several studies mention that the performance of CAMSRA differs depending on the region (Wang et al., 2020; Wagner et al., 2021).

For instance, it has been shown that there is more agreement between CAMSRA and observations over Europe than in the Tropics (Errera et al., 2021). CAMS also provides daily forecasts (hereafter CAMSFC), which have a finer horizontal resolution and a larger number of vertical model levels than CAMSRA. System upgrades and verifications of CAMSFC are reported in several studies (Schulz et al., 2021; Eskes et al., 2021). A recent validation based on various observations shows that, in terms of bias, CAMSFC overestimates surface ozone values at most of the stations (Sudarchikova et al., 2021).

However, it shows significant correlations across most of the stations, e.g., in China.

Despite many evaluation studies of CAMSRA and CAMSFC in different parts of the globe, less attention has been given so far to Iran, which is a country with a complex topography and diverse meteorological systems that contribute to the ozone

levels in this area. This study aims to address two questions: (1) how are the performances of CAMSRA and CAMSFC in simulating ozone over this region? (2) To what extent can downscaled CAMS datasets be used to study surface ozone at a city scale? To compensate for the limited spatial resolutions of the models, we downscale the CAMS ozone using the Long Short-Term Memory (LSTM) technique. The data are compared with the measured ozone data at 27 air quality monitoring stations distributed over different parts of the country. That allows us to assess the CAMS over diverse zones, e.g., a highly populous and polluted area vs. a small and desert-like town.

A detailed description of the datasets used in this study is presented in Sect. 2. The methodology is explained in Sect. 3, and the results are shown in Sect. 4. The discussion is presented in Sect. 5, and the paper ends with the conclusion's remarks in Sect. 6.

## 2 Description of Data

### 2.1 CAMS products

This study uses two data products, namely CAMSRA and CAMSFC, that have been produced by the ECMWF in the framework of the CAMS. These datasets focusing on surface ozone are introduced in the following subsections. An overview of the main differences and similarities between both products is given in Table 1. For more details on other aspects, the reader is referred to the references.

### 2.1.1 CAMS reanalysis (CAMSRA)

This product is the latest (state-of-the-art) global CAMS reanalysis dataset of atmospheric compositions. They are produced using a four-dimensional variational (4D-Var) scheme as an assimilation technique. The chemistry module of the CAMS relies on the IFS (CB05) tropospheric chemistry mechanism with 52 species and 130 reactions (Huijnen et al., 2010; Flemming et al., 2015; Huijnen et al., 2020). Dry deposition velocities are derived from the SUMO model (Michou et al., 2004). Anthropogenic emissions are based on the MACCity inventory (Granier et al., 2011), with modified wintertime CO emissions over North America and Europe (Stein et al., 2014). Monthly mean biogenic VOC emissions are derived offline from MEGAN (Guenther et al., 2006), using NASA's Modern-Era Retrospective Analysis for Research and Applications (MERRA) reanalyzed meteorological fields (Sindelarova et al., 2014). Daily biomass-burning emissions originating from the Global Fire Assimilation System, version 1.2 (GFASv1.2, Kaiser et al., 2012) are inferred from satellite observations of fire activities. The meteorological model consists of the given version of the Integrated Forecast System (IFS), i.e., CY42R1, with an interactive ozone and aerosol radiation scheme. Comparing to the previous atmospheric chemistry CAMS reanalysis data, CAMSRA has a finer horizontal resolution of 80 km with 60 vertical model levels, with the top level at 0.1 hPa. CAMSRA covers data for

the period of January 2003 to December 2021. The data are archived in 3 hourly time intervals. Hereafter, the ozone from this dataset is called $O_3^{RA}$.

**2.1.2 CAMS forecast (CAMSFC)**

In addition to the aforementioned datasets, CAMSFC issues (and produces) a daily global forecast of atmospheric compositions

twice a day, which is initialized from analysis at 00:00 and 12:00 UTC. The forecast consists of more than 50 chemical and seven different aerosols, providing also several meteorological parameters. Compared to CAMSRA, in CAMSFC the initial conditions of each forecast are obtained from analysis of atmospheric composition in near-real time, i.e., combining the previous forecasts with satellite observations using the 4D-VAR data assimilation technique. CAMSFC uses an atmospheric model to determine the evolution of the concentration of all species over time for the next five days. Apart from the required

initial state, it also uses inventory-based or observation-based emissions estimates as boundary conditions at the surface. Biogenic emissions originate from CAMS-GLOB-BIO v1.1, which is calculated from the MEGAN v2.1 model using ERA-Interim meteorology (Sindelarova et al., 2022). Monthly average of anthropogenic emissions is derived from the CAMS_GLOB_ANT v2.1 inventory based on a combination of EDGAR v4.3.2x and CEDS emissions (Granier et al., 2019). Biomass burning emissions are based on GFAS. Dry depositions of trace gases are calculated online. Sulfur species, nitrate,

and ammonium are coupled between chemistry and aerosol schemes. In contrast to the CAMSRA, CAMSFC is available at a finer horizontal resolution of 40 km. CAMSFC is upgraded regularly, e.g., once a year, during which the model's resolution can change or new species can be added. From 9 July 2019 onwards, CAMSFC uses the assimilation system's IFS CY46R1, in which the vertical model levels have been upgraded from 60 to 137. Details of other upgrades to this system can be found in Haiden et al. (2019) and Basart et al. (2019). IFS CY47R1 was used on 6 October 2020, with some upgrades in observations,

emissions, and model changes (Eskes et al., 2021; Sudarchikova et al., 2021). The temporal coverage of the CAMSFC is from 2015 to the present, with temporal resolutions of 1 hourly (only for surface fields) and 3 hourly. This study uses 3 hourly forecast fields from 00:00 UTC up to 24 hours. Hereafter, the ozone from this dataset is called $O_3^{FC}$.

**2.2 In situ measurement datasets**

Surface-based measurements of ozone were extracted from the Tehran air quality control portal, which is publicly available,

for 21 stations. A couple of the stations contain no data records, and the data sparsity at the stations differs from year to year. Hourly time series of surface ozone for other cities are not accessible to the public and were obtained from the Iranian Environmental Protection Organization for 54 air quality monitoring stations. We added the Geophysics station, which is located at the Geophysics Institute, University of Tehran, Tehran. This station measures surface ozone along with several other variables such as air temperature, nitrogen oxides, wind, total ozone column, etc. Most of the air quality monitoring stations

in Iran are installed in the cities, as they are aimed for the public health report. There is no information about stations' type or availability of the data at background sites. To have a common quality, the validity of the data was checked by performing a

few statistical tests, such as (1) range test: verifies if the values are within the acceptable range limits (Zahumensky, 2004; Taylor and Loescher, 2013); (2) constant value test: checks the required variability among successive values (Zahumensky, 2004); and (3) discontinuity test: identifies suspicious data points before and ahead of the discontinuities (Zurbenko et al., 1996; Gerharz et al., 2011). We use the stations containing data for the year 2020, where more than 50 % of the data is available for each month. Table A1 lists the names and geographical locations of the stations, of which the first 22 are ordered based on the stations' latitudes. In Table A1, there is a number along with the stations' names, hereafter, the stations are referred with these numbers. To include more stations in the analysis, we consider five more stations in Table A1, i.e., from 23 to 27, which only one or two months of the year 2020 contain less than 50 percent of data (see Figure A1). The distribution of the stations is shown in Fig. 1, which covers three large cities (Tehran, Shiraz, and Tabriz) and six small cities (Birjand, Gilan, Hamedan, Zanjan, Markazi, and Yazd). Hereafter, the observation datasets and observed ozone time series are called OBS and $O_3^{OBS}$, respectively.

Both reanalysis and forecast datasets were co-located with OBS through temporal and spatial interpolations. OBS data are available in hourly resolution, in contrast to the CAMS datasets that are available in 3 hourly intervals. To match the frequency of the CAMS outputs with OBS, 3 hourly observed values are considered in such a way that at least two hourly values are available; otherwise, it renders the value as missing.

## 3 Methodology

This section has been divided into three sections. Sect. 3.1 details the theory of decompositions and the method used in this study. Sect. 3.2 describes the procedure for neural network modeling and the pre-processing of its input. Sect. 3.3 defines the metrics (indicators) that have been used to assess the CAMS performance and error sources.

### 3.1 Spectral decomposition of the time series

The presence of various scales of motion, which are caused by several physical and chemical processes, in the time series of $O_3$ can complicate the analysis and interpretation of data. As an example, short-term and fast fluctuations in the $O_3$ time series are majorly attributed to chemical processes such as NO titration, whereas long-term and seasonal variations are mainly related to solar radiation, long-range transport, and stratosphere-troposphere exchange (Monks, 2000). Scale analysis is a method by which the time series can be separated into different temporal terms. Here, the time series of $O_3$ is decomposed into three different spectral components, namely short (period less than 2 days), medium (period of 2–21 days), and long (periods longer than 21 days) terms, by applying the Kolmogorov-Zurbenko (KZ) technique (Rao et al., 1997). KZ is essentially a low-pass filter that consists of repeated moving averages. Its use has been demonstrated in earlier studies (Hogrefe et al., 2000; Kang et al., 2013; Seo et al., 2014). A detailed discussion of the KZ filter along with a comparison to other separation techniques can be found in Eskridge et al. (1997) and Loneck and Zurbenko (2020). KZ requires two input parameters, KZ (m, k), where $m$ is the window size for filtering and $k$ is the number of iterations. Since the values that have been commonly used for $m$ and $k$

in the literature may not be applicable for 3-hourly data, we selected them based on the criteria suggested in Yang and Zurbenko (2010):

$$m \times \sqrt{k} \leq p \tag{1}$$

KZ filters out all periods that are less than $p$, i.e., the number of filtered time intervals. Therefore, three components of interest in this study are estimated as follows:

$$S = O - KZ\ (5,\ 5) \tag{2}$$

$$M = KZ\ (5,\ 5) - KZ\ (35,\ 5) \tag{3}$$

$$L = KZ\ (35,\ 5), \tag{4}$$

where $O$ refers to the original time series and S, M, and L indicate the short, medium, and long terms, respectively. Here, the units of O and the spectral terms are in nmol mol$^{-1}$. As expected from Eq. (1), KZ (5, 5) filters all periods less than 11.2 time steps. This corresponds to 33.54 hours, or 1.4 days, as the data are recorded in an interval of 3 hours. The same holds for KZ (35, 5), which filters all periods less than 9.8 days. Hence, the S refers to the short scale fluctuations, which are done in less than 1.4 days. Similarly, M refers to synoptic scale events with time scales ranging from 1.4 to 9.8 days. The variations with the time scales of more than 9.8 days are represented in the L term.

### 3.2 Statistical downscaling

To bridge the spatial scaling issue between coarse resolution CAMS datasets and local-scale measured data, statistical downscaling (SD) methods have been developed (Wilby and Wigley, 1997). SD refers to the use of statistical-based techniques to determine a relationship between global scale models' outputs and observed local (small) scale variables (Wilby et al., 2004; Wilby and Dawson, 2013). There are numerous SD methods such as linear regression (Sachindra et al., 2013; Beecham et al., 2014), stochastic weather generators (Wilks, 1999; Kilsby et al., 2007; Semenov and Stratonovitch, 2010), and artificial neural networks (Tripathi et al., 2006; Ahmed et al., 2015; Sachindra et al., 2018; Sebbar et al., 2023), to name a few. In this study, a deep learning method known as the LSTM network was used to analyze the complex relationship between $O_3$ and its precursors. LSTM is a modified version of a recurrent neural network designed to handle long-term (and short-term) dependencies in sequential data (Hochreiter and Schmidhuber 1997). LSTM contains memory cells that can hold (and store) information for a long time, so making them suitable for time series analysis. The standard LSTM consists of three gates: input, forget, and output gates for controlling the movement of information. We use Keras, a high-level neural network Python library ("Keras: the Python Deep Learning library", Chollet, 2015; https://keras.io) to build and train the LSTM model. This model requires a specific configuration and tuning to work effectively with the datasets. A range of control values for several hyperparameters (Table A2) were tested by multiple trial-and-errors. The most effective hyperparameters (Table A5) were selected using the Random Search optimization method.

To prepare the LSTM inputs, several meteorological variables (Table A3) were obtained from the CAMSRA and CAMSFC datasets. To prevent overfitting of the model, a cross-validation Lasso regression was performed to identify the potential

predictors at each station. The lagged $O_3$ (from OBS) was also considered as the model inputs, since the concentration of $O_3$ is not only affected by meteorological factors but also by the influence of the $O_3$ levels in the past. A partial autocorrection function was utilized to estimate the correlation between observed $O_3$ at time T and earlier time steps. For most of the stations, the autocorrelation coefficients decrease after a time lag of 24 hours within a confidence interval of 95 %. So, the $O_3^{OBS}$ at times T-1,…, and T-8, were also considered predictors at each station. Selected predictors and observed $O_3$ were decomposed using Eq. (2), (3), and (4).

In order to provide the final output, i.e., downscaled $O_3$, the LSTM architecture was trained on the decomposed datasets. The data records were divided into 65 % for the training subset and 35 % for the validation subset. The best model was chosen based on the $R^2$ score (coefficient of determination) and mean square error (MSE). The selected model was applied to all data records to provide a downscaled output. All these procedures were applied to each station separately and are illustrated in Fig. 2.

## 3.3 Model evaluation

We use the mean square error (MSE) as a metric to evaluate the models' performance. The MSE is defined as the squared mean of the difference between modelled ($x_m$) and observed ($x_o$) variables.

This metric can be modified to include all relevant model evaluation indicators, i.e., bias, variance, and correlation, as (Murphy, 1988; Solazzo and Galmarini, 2016):

$$MSE = (\bar{x}_m - \bar{x}_o)^2 + (\sigma_m - r\,\sigma_o)^2 + \sigma_o^2(1 - r^2) \tag{5}$$

where $\sigma_m$ and $\sigma_o$ refer to the standard deviation of the modelled and observed data, respectively, and $r$ is the coefficient of correlation between the observed and assimilated datasets. In Eq. (5), the first term (hereafter E1) shows the deviation between average modelled ($\bar{x}_m$) and measured ($\bar{x}_o$) datasets and refers to the model accuracy. The second term (hereafter E2) contains the variance error, i.e., the discrepancy in amplitude or phase between the variability of the modelled and observed values, that determines the precision of the model. Also, the third part (hereafter E3) refers to unsystematic errors related to the associativity between observed and assimilated datasets. In other words, the E2 indicates an explained error, which reveals the variance error arising from the variability of the modelled variables that are not observed in measurements. That could arise from overfitting associated with complex chemical processes in the model or imbalance among coupled components. The E3 represents an unexplained error, reflecting the lack of observed variability in the modelled data. That refers to the variabilities which are not captured by the models, even though those variabilities exist in the observations. The E3 can arise from random and non-representative errors caused by sub-scales and non-resolvable processes in the observations, or from a deficiency of the model in capturing meso-scale phenomena. Due to the spectral decomposition of the data, the S and M components have zero mean fluctuations. Hence, the E1 term in Eq. (5) is zero, and only the E2 and E3 terms are analyzed below.

To compare the distribution of error of modeled $O_3$ before and after downscaling, the skill score (SS) is calculated as (Wilks, 2006)

$$SS = 1 - \frac{MSE}{MSE_{ref}} \qquad (6)$$

Here, $MSE_{ref}$ and MSE refer to the MSE of $O_3^{RA}$ (or $O_3^{FC}$) and downscaled $O_3$ ($O_3^{SD}$), respectively. The value of SS varies between 0 and 1. The value is zero once there is no preference in $O_3^{SD}$ with respect to $O_3^{RA}$ (or $O_3^{FC}$), i.e., the $O_3$ variability is not explained by selected predictors. The value of SS is one when the MSE of $O_3^{SD}$ is zero, which means the whole $O_3$ variability in the LSTM model is explained by the predictors, i.e., the LSTM model is perfect.

## 4 Results

### 4.1 Spectral components

The time series of $O_3$ and all meteorological variables for OBS and CAMS datasets decompose into three spectral components, short (S), medium (M), and long (L), by applying the method (KZ filter) explained in Sect. 3.1. Figure 3 shows the original time series of $O_3^{OBS}$, $O_3^{RA}$, and $O_3^{FC}$ and their estimated spectral components at the first station. To clearly see the signals, we only show part of the time series, here for the summer months (June, July, and August: JJA). Looking at the original 3 hourly time series (Fig. 3a), both CAMS datasets overestimate and underestimate ozone during different periods, but it is difficult to determine any clear patterns or identify specific reasons for the model bias. The S component contains frequent fast oscillations occurring every day with regular maxima and minima (see Fig. 3b). In this figure, the amplitude of S oscillations of the $O_3^{RA}$ and $O_3^{FC}$ is different from that in OBS, indicating differences in the diurnal cycle of observed and simulated ozone mixing ratios. The M term captures variability on the timescale of synoptic systems. Some episodic events are more visible in the M component than in the S component. For instance, in Fig. 3c, the M component of the OBS represents a clear signal of an episodic event in the middle of June. This episode is not well captured in CAMSRA while it is captured in CAMSFC. It seems that for most of the periods, the variations of the M component in both CAMS datasets are in good agreement with those in OBS, while the amplitudes of oscillations in CAMS do not correspond well with those in OBS. The underestimation and overestimation of the amplitude (with respect to observations) in CAMSFC is less than that in CAMSRA. Compared to the S and M terms, which oscillate around zero, the mean values of the L components are not zero (see Fig. 3d). The L represents variations of the ozone mixing ratios on seasonal, semi-seasonal, and multiannual timescales. Comparing the variations of CAMSRA and CAMSFC with OBS for L shows more similarity between CAMSFC and OBS than between CAMSRA and OBS. Both models exhibit a high bias with respect to the ozone mixing ratios. Nevertheless, the decomposition of the L component is not reliable due to the limited period (one year) of the available data, so hereafter we only assess the S and M components.

### 4.2 Variable selections

The time series for 16 relevant meteorological variables were extracted from CAMS products. To avoid model overfitting, we identify potential predictors of the variables. To decide on the importance of the variables, we used the LassoCV estimator.

The relationships between predictors and $O_3^{OBS}$ were estimated by performing LASSO (Least Absolute Shrinkage and Selection Operator) regression. The variables with the highest absolute Lasso coefficient (importance weight) are considered the most important. Figure. 4 shows that the T2m is the most explanatory meteorological variable and NO, $NO_2$, and $O_3^{RA}$ are the main chemical variables for CAMSRA_S at most of the stations. The variables with high feature importance (weight > 0.1) were considered for use in the LSTM modeling. Table 2 lists selected predictors for both components of CAMSRA at each station. At station 1, twelve variables, namely, T, V, U10m, V10m, MSLP, SP, T2m, SH, W, CO, $NO_2$, and $O_3^{RA}$, are identified as the potential predictors of the S component, while four variables, i.e., U10m, W, $SO_2$, and $O_3^{RA}$, are selected for the M term. Some of the selected predictors are common between the S and M components. A few meteorological variables such as T2m, SP, MSLP, W, and U10m (or V10m) appear for the S component at most of the stations. These variables reflect the information about temperature, pressure, and vertical velocity. Temperature is one of the key meteorological factors influencing the S term variability of $O_3$ through its effect on biogenic emissions, photochemical kinetics reaction rate, and anthropogenic emissions. Stable anticyclones and sunny conditions promote $O_3$ formation and accumulation. Zonal and meridional winds at 10 meters are important for the dispersion of ozone precursors at local scales. For most of the stations, the S term is affected by pollutant species such as $O_3^{RA}$, NO, and $NO_2$, of which NO and $NO_2$ are recognized as potential drivers of $O_3$ levels. Selection of TCC and FCC for the M component at most stations indicates that cloud covers are mostly associated with synoptic systems (e.g., occurrence of high pressure systems associated with clear-sky conditions) and $O_3$ variability on this scale. The M component at a few stations, e.g., 4, 6, 9, 13, etc., shows weak associations with the parameters, so no variables are selected for them. This situation often happens for the M component and suggests the role of other factors (not included in the predictors). There are a few stations where $O_3^{RA}$ ($O_3^{FC}$) is not selected as an important variable, which is related to the small (weak) associations between $O_3^{RA}$ ($O_3^{FC}$) and $O_3^{OBS}$. For instance, SH is selected as the main factor effecting the M term at station 23, i.e., Rasht. This station is located between the mountains (Alborz) and coast (Caspian Sea), with a local environment of rainy with a humid subtropical climate. That is similar to the Western Mediterranean regions, where a lack of strong synoptic advection, combined with the orographic characteristics and the land-sea breezes, favors episodes of high ozone levels over this region (Millan et al., 1999; Velchev et al., 2011; Wentworth et al., 2015). Similar to the CAMSRA, for the CAMSFC the number of selected parameters for the S is larger than that for the M (see Table A4). In CAMSFC, BLH and V10m (or U10m) appear as dominant meteorological drivers affecting the S component. Stable boundary layer height causes the accumulation of ozone and its precursors during night or under light (weak) winds conditions. Moreover, ozone in residual layer can be transported over long distances with prevailing winds. In the morning, trapped ozone can be entrained downward into the mixed layer (Stull, 1988; Zhang and Rao., 1999). The M term is mostly associated with $O_3^{FC}$.

## 4.3 LSTM model and validation

The LSTM model was trained and validated with the datasets, as explained in Sect. 3.2. We tuned hyperparameters, which allow the learning algorithm to run until the error from the model, i.e., the loss function, has been sufficiently minimized. As

there are no given values to set these numbers, the optimum values were obtained by multiple trial-and-error tests (see Table A5). The best model was selected based on MSE and $R^2$ score (coefficient of determination), which indicates the amount of explained variance by the LSTM model. Figure 5 shows the $R^2$ of the selected model for all data series at each station. For most of the datasets, the $R^2$ is larger than 0.5, indicating that more than 50 % of the $O_3$ variance is explained by the LSTM. The $R^2$ for the M component is larger than that for the S term, despite the smaller number of predictors for the M. That might reflect that the M component is easier to be modeled due to less complexity. In this figure, the $R^2$ of the M is around 0.9 for all stations, while it varies for the S term. The $R^2$ value of the S at the stations over the city of Tehran is within the same range of 0.7 to 0.8. Both CAMSRA and CAMSFC show the $R^2$ less than 0.5 for the S term at a few stations, namely 22 (Yazd), 24 (Zanjan), and 25 (Markazi). A possible reason for that could be the peculiar characteristics of short-term ozone variability at these sites or their geographical locations. Model to model differences in $R^2$ are more pronounced in the S, that is likely due to the different emissions inventories used in the models.

Figure 6 shows the box plots of MSE and different terms of MSE, i.e., E2, E3, for both components of $O_3{}^{SD}$. For the sake of simplicity, descriptions of the results are mostly based on the mean values. Nevertheless, the values of the indicators at each station are shown as a scatter point next to the box plots. From Fig. 6a, it turns out that the mean MSE (shown with red squares) of $O_3$ for the S component is larger than that for the M component for both models. That could arise from the uncertainties in $O_3$ precursor emissions affecting modeled local photochemistry and likely S variability. The largest value of the MSE is associated with the $O_3{}^{SD}$ of the stations located in the city of Tehran. That can be associated with the uncertainties in CAMS emissions inventories, which may have larger impact in cites with high anthropogenic emissions sources. The stations in the northern part of the city (e.g., stations 4, 5, 6, 7, 8, and 9) show a larger MSE than the stations in the southern part (e.g., stations 10, 11, 14, 15, 16, 17, and 19). That can be associated with the deficiency of the emissions inventories in capturing the local emissions changes within urban areas. The large value of MSE is also found for the S term at the stations located in Shiraz and Tabriz, which are known as big and highly populated cities with numerous local anthropogenic emissions sources (e.g., thermal power plants, oil refinery, cars, etc.). Station 2 in Tabriz shows less MSE than stations 1 and 3, which are located in the industrialized part of the city. That can be associated with the uncertainties in the spatial variations of the emissions inventories used in CAMS. Although the CAMS anthropogenic emission inventories account for emissions from different sectors, such as transportation, residential and energy sectors, as well as biogenic fluxes, they have a temporal and spatial allocation with a monthly spatial grid resolution of 0.1°x0.1°. Low values of the MSE for CAMSRA_S and CAMSFC_S are attributed to stations 22 (Yazd), 20 (Hamedan), and 24 (Zanjan). Similar to $R^2$, the lowest MSE belongs to the Yazd station, which contains fewer local emissions sources than other cities such as Tehran, Tabriz, and Shiraz.

Fig. 6b shows the explained error (E2) in CAMSRA and CAMSFC for both components. E2 is a model related error, a possible source for this can be a misrepresentation of short- and meso-scale phenomena in models. The small values of E2 reflect the low contributions of E2 to the MSE and the noticeable improvement of the $O_3{}^{SD}$ (via downscaling procedures). The major portions of the MSE are associated with the unexplained errors (E3) for both components, see Fig. 6c. The E3 for the S component is larger than that for M, as expected from the variance of these components. The S variability is associated with

the effect of daytime photochemical production, downward transport of $O_3$ rich from upper levels, combined with $O_3$ loss by depositions (in the surface layer). A large value of E3 for the S component can arise from the CAMS' deficiency in resolving the meso-scale phenomena such as local winds, NO titration, deposition rates and their influences on $O_3$ variability. Assessing the element of E3 (see the third term of Eq. (5)) shows that large variances of observations ($\sigma_o$) or small correlations ($r$) cause the large E3 and consequently the large MSE. Fig. A2a shows the correlation between the models and observation datasets for both components. This figure shows that M contains a larger correlation ($r > 0.9$) than S in both models. A high value of correlation between two terms can be attributed to the larger covariance of two terms or the less variance of each term. Fig. A2b shows the covariance between models and observations. As can be seen in this figure, the mean value of covariance for the S components is larger than the M. So, the smaller correlation of S in comparison to that of M is attributed to the larger variance of S (Fig. A2c). In other words, the better model performance (i.e., smaller E3 and MSE) for the M is not associated with the larger covariance of the M component. That is attributed to the less variance of the M than that of the S, see Fig. 3 and Fig. A2c.

In order to examine the effect of the CAMS products and lagged $O_3$ (from actual observations) on the LSTM model, we exclude the measured lagged ozone from the predictors of the LSTM model, hereafter LSTM$^{no\_lag}$. The $R^2$ of the LSTM$^{no\_lag}$ is shown in Fig. A3. Overall, the $R^2$ of the LSTM$^{no\_lag}$ is less than that of the LSTM. This suggests that the LSTM$^{no\_lag}$ may carry the risk of not including all important predictors (e.g., lagged ozone) in the model. This feature is more noticeable in the M term than the S term, i.e., the $R^2$ of the S component is less affected by removing the lagged $O_3$. That reflects the CAMS products, which explain more of the S variance than that of the M term. In other words, most of the variance of the M term in the LSTM is explained by the lagged $O_3$ (not by the CAMS products). That could be a reason for the better performance (less MSE) of the M than the S. Figure A4a shows the MSE of the LSTM$^{no\_lag}$. In this figure, the MSE of the datasets increases by two times with respect to that of the LSTM. The higher values of the MSE in the LSTM$^{no\_lag}$ are attributed to the removal of the observed lagged $O_3$ from the model. Although the $R^2$ of the LSTM$^{no\_lag}$ for the S is larger than that for the M term, the MSE of the S is higher than that of the M term. This is similar to the MSE of the LSTM, which is related to the higher variance of S than M. Similar to the LSTM, in LSTM$^{no\_lag}$, the low values of MSE are seen for the S component of $O_3$ at stations 22 (Yazd), 20 (Hamedan), and 24 (Zanjan).

The skill score (SS) of the downscaled models $O_3^{SD}$ with respect to the $O_3^{RA}$ and $O_3^{FC}$ for all datasets are shown in Fig. 7. In panel (a) of this figure, the mean value of SS for three datasets, namely CAMSRA_S, CAMSRA_M, and CAMSFC_M is larger than 0.9. This reflects that the downscaling procedure (LSTM) improves the accuracy of the results in the three mentioned datasets. The lower value of the SS for CAMSFC_S can be attributed to the higher skill of the reference dataset, i.e., $O_3^{FC}$, or the less accuracy of the LSTM model. The SS of the LSTM$^{no\_lag}$ for CAMSRA_S shows the same high accuracy as that in the LSTM, whereas for other datasets the mean SS declines to less than 0.8 (see Fig. 7b). There is a large difference between SS of the LSTM and LSTM$^{no\_lag}$ for the M component, which shows the importance of the lagged $O_3$ for modeling of the M term. Larger values of SS for the CAMSRA than that for the CAMSFC reflect a better performance of $O_3^{FC}$ over Iran, That is also shown in Fig. A7a, in which the MSE of CAMSFC_S is less than that of CAMSRA_S.

**5 Discussion**

Analysis of the spectral components in this study reveals that the $O_3$ variability in both CAMS products possesses a nearly similar shape (although in different phases and amplitudes) as those in OBS. Although both datasets share many of the same
parameters in common, there are several differences that distinguish $O_3^{RA}$ from $O_3^{FC}$. $O_3^{FC}$ is produced by a model with finer horizontal and vertical resolutions. Different anthropogenic and biogenic emissions have been used in both models (see Table 1). CAMS-GLOB-ANT (used in CAMSFC) provides up-to-date emissions of air pollutants and greenhouse gases, at the spatial and temporal resolution required by the model (0.1°x0.1°). CAMSRA uses MACCity emission inventory with a resolution of 0.5°x0.5°. Figure A6 shows a comparison of CAMS-GLOB-ANT and MACCity for a couple of ozone precursors, i.e., $NO_x$
and CO. Compared to CAMS-GLOB-ANT, MACCity provides higher $NO_x$ and CO emissions. CAMS-GLOB-ANT shows more details of the emissions' variability due to the finer spatial resolution. The area with the highest emissions in both inventories are located over Tehran.

The results of the models' performances show a larger MSE for the S than that for the M in both CAMS. That arises from the larger variance of the S in comparison to the M (Hogrefe et al., 2000; Hogrefe et al., 2014; Kaffashzadeh 2018; Kaffashzadeh
and Aliakbari Bidokhti, 2022). The results of the error apportionment show the negligible contribution of the E2 to the MSE. E2 arises from the limited spatial resolutions of the CAMS in capturing short- and meso-scale phenomena that are attenuated (alleviated) by the SD procedures. The MSE has mostly arisen from the E3, which emphasizes the lack of observed variability in the CAMS data. The E3 assessment shows less variability for both components of $O_3^{SD}$ than in $O_3^{OBS}$. That could arise from random errors inherent in the OBS data due to sub-scale or non-resolvable processes in an observational network. The
variability in the measured data might be generated from the non-representatives' errors due to random effects caused by turbulence or sub-scale perturbations (Gandin, 1988; Steinacker et al., 2011). It is not straightforward to distinguish and exclude these errors in the measured data because of their chaotic and unsystematic behaviors. Adding the lagged $O_3$ to the predictors of the downscaled model halfs the E3 (and MSE). Less MSE of the M in comparison to that of the S attributes to not only the less variance of the M than the S but also the larger contribution of the lagged $O_3$ in the M than that in the S (as
shown in Sect. 4). The S component shows large associations with meteorological variables such as T2m, BLH, U10m, and V10m and pollutant species such as CO, NO, and $NO_2$. That is due to short-term $O_3$ fluctuations associated with processes such as vertical mixing, local NO titration, depositions, wind speeds, solar flux, etc.

The S component shows the large value of MSE for the stations located in Tehran, Shiraz, and Tabriz, which are known as the most populated cities (and so large local emissions sources) in Iran. The largest MSE belongs to $O_3$ at the stations over Tehran
(see Fig. A4). That can be partly attributed to the complex topography and local (meso) scale flow (e.g., slope, mountain, and valley flow) over the city. The pollutant concentrations are highly affected by these factors, which are hardly captured by the global chemistry models (Fiore et al., 2003). The MSE of $O_3$ over Tehran in the warm season is much higher than that in the cold season (see Fig. A5). That could arise from the uncertainty of $O_3$ precursors in CAMS, as they are not adjusted by data assimilation systems. CAMS-GLOB-BIO (used in CAMSFC, see Table 1) provides a monthly average of the global biogenic

emissions, which are calculated using the MEGAN (used in CAMSRA, see Table 1), driven by ERA-Interim meteorological fields. In summer, rising temperatures speed up the rate of many reactions and enhance biogenic VOC emissions (Sillman and Samson, 1995). The city of Tehran suffers from high levels of emitted $NO_x$ from several sources, such as road traffic, industrial activities, the energy conversion sector, etc. (Hosseini and Shahbazi, 2016; Yousefian et al., 2020). The latest Tehran emission inventory indicates that the annual emissions of VOC and $NO_x$ are approximately 91 and 103 thousand tons, respectively

(Shahbazi et al., 2022). The contributions of vehicles to VOCs and $NO_x$ emissions are estimated to be 79 % and 35.2 %, respectively, and increase to 79.5 % and 37.2 %, respectively, in the summer. In addition to the aforementioned factors, what distinguishes Tehran from other cities is the difference between day and nighttime populations. During the day, traffic in Tehran reaches its highest level due to the arrival of private vehicles and passenger and cargo transportation vehicles from surrounding areas and cities. This issue has a significant impact on the city's traffic and the vehicle traffic on intercity routes

leading to Tehran. The impact of these (meso scale) factors cannot be captured in a global emissions inventory with a limited resolution. That induces large model uncertainties, in particular for the S variability, which has large associations with pollutant species. Besides, for some periods the emissions are not available and so prescribed, which means they are either kept fixed since the last year available or extrapolated (projected) with a climatological trend. MACCity emission inventory has not been updated since 2010, and recent years are only based on projections of past trends. CAMS-GLOB-ANT provides the monthly

average of the global emissions of 36 compounds over the period 2000-2019. The MSE distribution over Tehran is uneven; the northern part of the city shows a larger MSE than that over the southern part. That can be attributed to the uncertainty of the simulated CO species, as it is selected as a predictor at the stations located in the northern part. The CO concentration increases, moving from the south to the north of Tehran (Sharipour and Aliakbari Bidokhti, 2014).

       Stratospheric ozone can affect surface ozone levels indirectly through vertical downward transport of ozone from the lower

stratosphere and/or the upper troposphere in larger time scales (Zanis et al., 2014; Akritidis et al., 2016) or directly through intense stratospheric intrusions (rarer) (Akritidis et al., 2010; Chen et al., 2022). Over Tehran, a major portion of $O_3$ during spring is transferred from the stratosphere (Aliakbari Bidokhti and Shariepour, 2007). A study by (Shariepour and Aliakbari Bidokhti, 2013) showed that several mid-latitude low pressure weather systems accompanied by tropopause folding affect northern Iran (Caspian Sea), and can cause downward transport of stratospheric ozone rich air towards the surface. During

summer, the occurrence of tropopause folding and their intensity over the Eastern Mediterranean and the Middle East regions are majorly controlled by the Asian monsoon. Since the zone of upper level baroclinicity and fold occurrences spread northwestward over this region, it first reaches Iran in July (Tyrlis et al., 2014). The large MSE of $O_3^{SD}$ for the cities of Shiraz and Tabriz is mostly associated with the geographical locations of the cities. Tabriz is the largest economic (industrialized) hub and metropolitan area in northwestern Iran, which is often affected by cyclonic activities (Asakereh and Khojasteh, 2021)

and summer circulations over the Eastern Mediterranean region (Tyrlis et al., 2013). Although CAMSRA captures the long-range transport processes and atmospheric background in the troposphere, it shows a lower skill over the Mediterranean, in particular the eastern part, compared to other regions (Errera et al., 2021). Shiraz, as the capital of Fars province, is the largest city with more than 1.2 million inhabitants in the southwestern of Iran. This city has high levels of air pollutions due to

population growth, urbanization, and traffic-related emissions. The city is located in a valley between two mountain ranges with east-west orientations. The model representation of the terrain is considered to be an important key factor for achieving a good representation of the wind flow in complex terrain (Mughal et al., 2017). The low MSE values in the cities of Yazd, Hamedan, and Zanjan are associated with the station locations, which are less populated and affected by the emissions sources. To assess the sensitivity and robustness of the results to SD methods, the data are downscaled using another SD method, namely the multiple linear regression (MLR) model. In this model, the predictors and predictand were the same as the LSTM model. Figure A7b shows the MSE of $O_3^{SD}$ with the MLR model. In similarity with the LSTM model (similar to the results in Sect. 4), the MSE for the S is larger than the M components downscaled with the MLR model. Although the mean value of the MSE of the downscaled data with the MLR is slightly larger than that of the LSTM. That could arise from the larger correlation (and covariance) between downscaled datasets and OBS in the LSTM model. Similar to the LSTM, the SS of the MLR is high for all downscaled datasets; the SS for the CAMSFC_S datasets is less than other datasets (see Fig. A8a). Two experiments were designed to assess the sensitivity of the model to less obvious predictors. In the first experiment, i.e., $MLR^{no\_lag(expr1)}$, the model was trained only using $O_3^{RA}$ and $O_3^{FC}$. In the second experiment, i.e., $MLR^{no\_lag(expr2)}$, the model was trained using the most influential meteorological variables (see Table A6). For the sake of simplicity (and being less expensive), both experiments were performed using the $MLR^{no\_lag}$ model. Table A7 lists the results of these experiments for station 22 (Yazd). As can be seen, the MSE of $MLR^{no\_lag(expr1)}$ and $MLR^{no\_lag(expr2)}$ are larger than that of $MLR^{no\_lag}$. That shows that part of the $O_3$ variability is explained by meteorology and partly by the chemistry ($O_3^{RA}$ or $O_3^{FC}$). Separating these two factors causes a decline of $r$ (see Fig. A9).

## 6 Conclusions

In this paper, the variability of $O_3$ in two datasets, namely CMASRA and CAMSFC, was assessed against observations at 27 urban stations distributed over Iran. Our observation datasets contain time series from various cities in Iran, e.g., highly polluted cities vs. small cities. This helps identify where the models capture reality and where they need more improvements. To cope with the limited spatial resolutions of CAMS, the data were downscaled using an LSTM neural network. The potential predictors (inputs) for the LSTM were identified from chemical and meteorological variables at each station. We decomposed all time series into three spectral components, i.e., short (S), medium (M), and long (L) terms. The S term consists of intraday and diurnal variations; the M term includes synoptic multiday fluctuations; and the other motions, i.e., seasonal, semi-seasonal, and trend, are carried in the L. We only assessed the S and M terms due to the availability of one-year data, i.e., 2020; the L component is primarily used to check the biases between model data and observations but should not be considered reliable with respect to trend analysis, etc. Since S and M components have zero-mean fluctuations, the bias term (the distance between the time average of model data and observations) is zero, and the main focus of this study was to analyze the variability terms, e.g., variance and covariance. The results presented in this study reveal several key points:

- Various variables were identified as potential predictors of ozone. The S term shows high associations with temperature, 10 m wind components, and $NO_x$, while the M component shows higher associations with cloud cover and simulated ozone. In CAMSFC, boundary layer height appears to be the dominant meteorological driver of the S component. The $R^2$ of the LSTM model for the M component is larger than that for the S term, despite a smaller number of predictors for M than for S. That might reflect that the M term is easier to be modeled.

- The SS of the downscaled CAMSFC_S is lower than other datasets. This can be attributed to the higher skill of the reference dataset, i.e., $O_3^{FC}$. The SS of the $LSTM^{no\_lag}$ for CAMSRA_S shows the same high accuracy as LSTM, whereas for other datasets, the mean SS declines to 0.5. That shows the importance of the observed (lagged) $O_3$ as a predictor in the LSTM. The robustness of the results was also confirmed using additional downscaling procedures, i.e., MLR.

- Both datasets, i.e., CAMSRA and CAMSFC, show less MSE for the M component than for the S term. That is mainly attributed to the low variance of M and is not related to the large covariance of this component. The MSE was mainly associated with unexplained model errors (E3), which could be caused by the CAMS deficiency in resolving the mesoscale phenomena such as local winds, NO titration, deposition rates, and their impact on $O_3$ variability. In addition, uncertainties in emission inventories might affect this error. Including a proxy of stratospheric ozone contribution to surface ozone (stratospheric ozone tracer) may be beneficial in explaining short term ozone variability, thus reducing the error (a recommendation for future work).

- In both datasets, the highest MSE appears for $O_3^{SD}$ at stations in the cities with high emissions, in particular over Tehran in the warm season. That majorly arises from the uncertainty of $O_3$ precursors, e.g., $NO_x$, in CAMS. This can be considered a starting point for improving the results of surface ozone, in particular at urban sites.

To date, most of the studies of ozone and other pollutants in Iran rely on reanalysis products, without using decompositions or downscaling procedures. Our findings show that the CAMSRA and CAMSFC datasets have some deficiencies in simulating ozone, in particular over the cities with high emissions of ozone precursors. Downscaling improves these products and makes them suitable for the study of ozone in major metropolitan areas. The method used in this study is not only applicable for the evaluation of the global models but also for prediction purposes.

**Code availability.** The Python 3.7 code of the methodology is available in the Zenodo (Kaffashzadeh, 2024).

**Data availability.** Part of the observational data is accessible from the http://airnow.tehran.ir/home/DataArchive.aspx portal. CAMS reanalysis and forecast datasets were obtained through ECMWF's atmospheric data service (last access: April 2022).

**Author contribution:** N. K designed the research, acquired and processed all data, performed the statistical analysis, and composed the figures and manuscript. A. A. B. contributed to proofreading.

**Competing interests:** The authors declare no competing interests.

**Disclaimer:** Publisher's note: Copernicus Publications remains neutral with regard to jurisdictional claims in published maps and institutional affiliations.

**Acknowledgements.** This study has been supported by the Center of International Science and Technology Cooperation
(CISTC). We thank the data providers of the Iran Environmental Protection Organization, Tehran Air Quality Control Company, and the ECMWF's climate data services. Appreciations is given to M. G. Schultz for his help in training N.K. Dr. F. Solmon is acknowledged for his valuable points during manuscript revisions. The authors highly acknowledge the constructive comments from two anonymous referees and topic editor.

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

**Table 1. An overview of similarities and differences between the CAMSRA and CAMSFC datasets used in this study**

| Name (references) | CAMSRA (Innes et al., 2019) | CAMSFC (Basart et al., 2019; Haiden et al., 2019; Sudarchikova et al., 2021) |
|---|---|---|
| Temporal coverage | 2003 to 2021 | 2015 to present |
| Assimilation system | IFS Cycle 42r1 4D-Var | IFS Cycle 46r1 (implemented on 9 July 2019) IFS Cycle 47r1 (implemented in 6 October 2020) |

| | | |
|---|---|---|
| Horizontal resolution | 0.75°x0.75° (T255) | 0.4°x0.4° (T511) |
| Vertical resolution | L60 Up to 0.1 hPa | L137 up to 0.01 hPa |
| Temporal resolution (output frequency) | 3 hourly | 1 hourly (surface level), 3 hourly (multi-level) |
| Anthropogenic emissions | MACCity | CAMS_GLOB_ANT v2.1 (cy46r1) CAMS_GLOB_ANT v4.2 (cy47r1) |
| Biomass burning emissions | GFASv1.2 | GFASv1.2 (cy46r1) GFASv1.4 (cy47r1) |
| Biogenic emissions | MEGAN | CAMS_GLOB_BIO v1.1 |
| Chemistry modules | modified CB05 | modified CB05 with a few upgrades such as dry depositions velocity, coupling with aerosol scheme, etc. |
| Input meteorological observations | As in ERA5 | As in ERA5 |

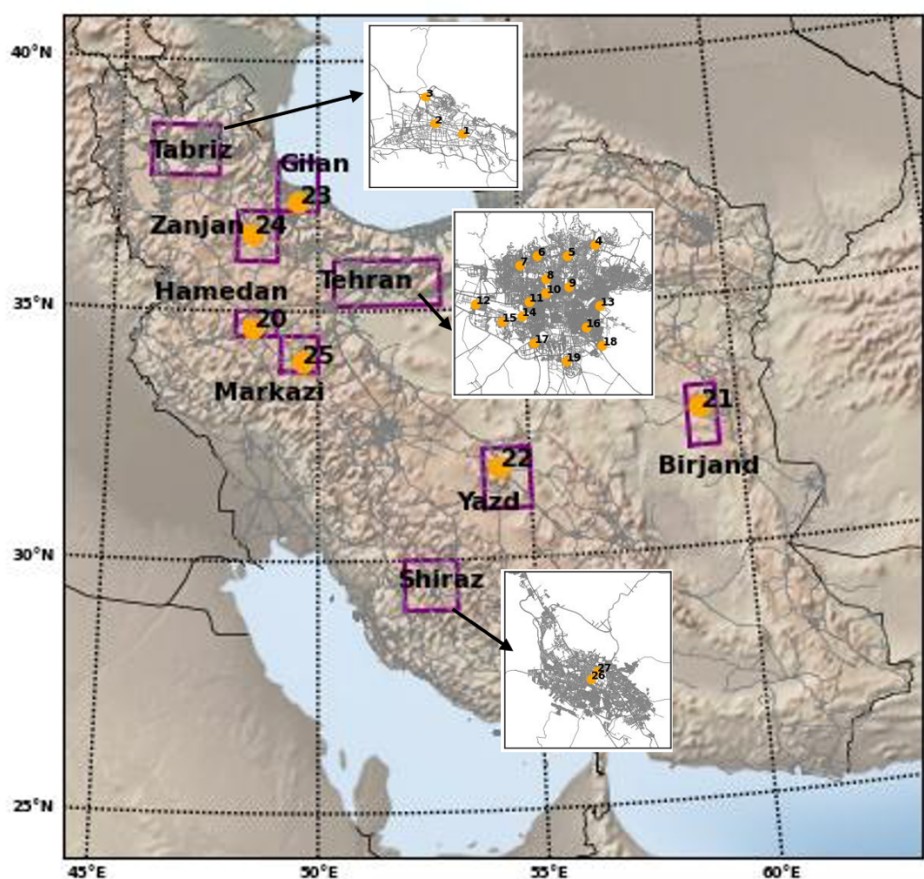

**Figure 1. Geographical location and distribution of the measured air quality stations used in this study. The purple box areas correspond to the locations of the cities. Here the stations are represented with a number, details on the name and geographical coordinates of the stations are given in Table A1. The arrows refer to the stations, which are overlaid on the cities' maps of Tabriz, Tehran, and Shiraz (Fars).**

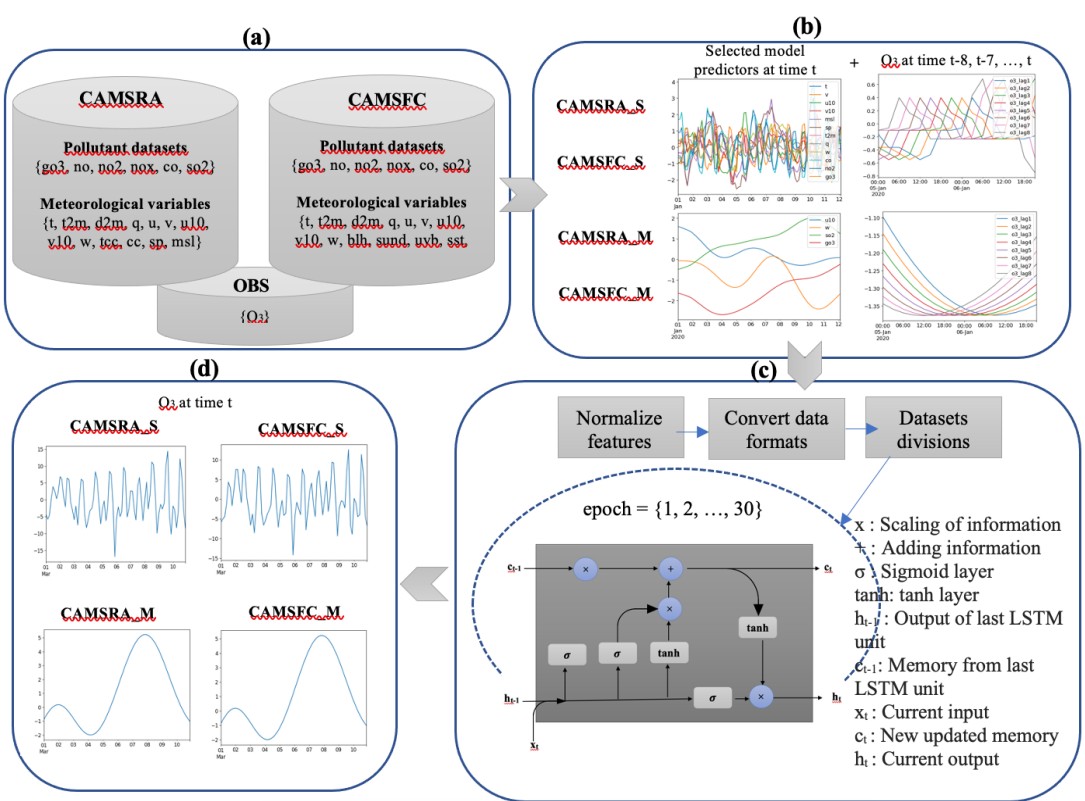

Figure 2. A schematic of the downscaling processes: (a) input data retrieval, (b) decomposition and prescreening, (c) LSTM modeling, and (d) downscaled datasets.

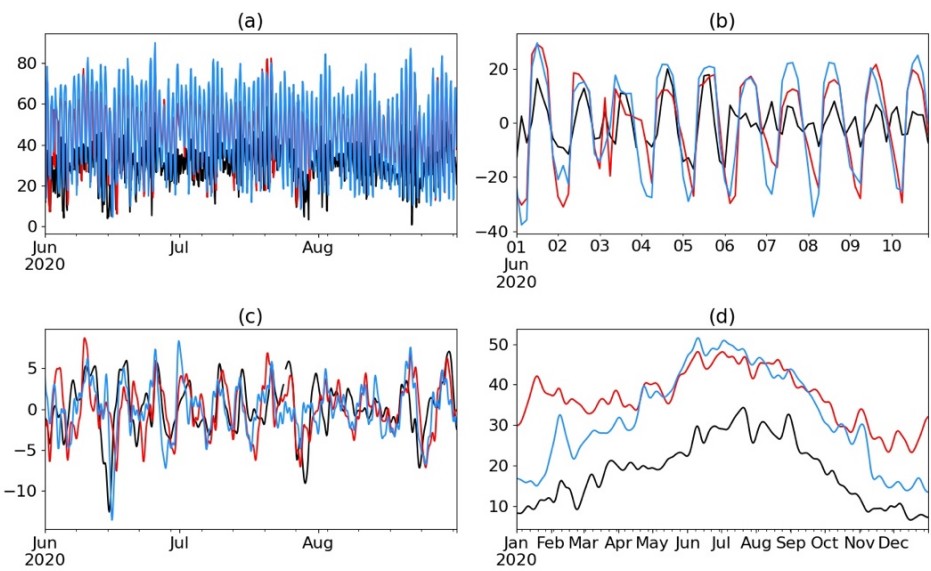

Figure 3. Different spectral components, i.e., (a) original time series, (b) short (S), (c) medium (M), and (d) long term (L) of $O_3^{OBS}$ (black), $O_3^{RA}$ (red), and $O_3^{FC}$ (blue) at station 1. The vertical axis in all panels shows the ozone mixing ratio in nmol mol$^{-1}$.

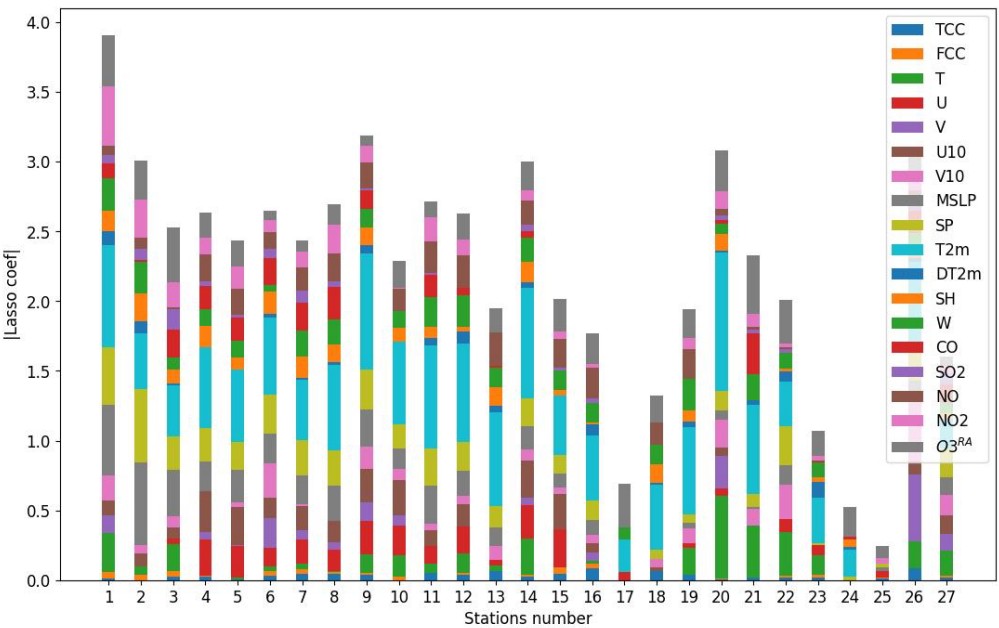

**Figure 4. Cross-validation Lasso regression to identify the potential predictors for ozone modeling. The higher absolute Lasso coefficient, the most important would be the variable.**

**Table 2. The most important explanatory variables of the CAMSRA at each station**

| Stations' number | S | M |
|---|---|---|
| 1 | T, V, U10m, V10m, MSLP, SP, T2m, SH, W, CO, NO$_2$, O$_3^{RA}$ | U10m, W, CO, O$_3^{RA}$ |
| 2 | MSLP, SP, T2m, SH, W, NO$_2$, O$_3^{RA}$ | U10m, SP, SO2, O$_3^{RA}$ |
| 3 | T, MSLP, SP, T2m, CO, SO$_2$, NO$_2$, O$_3^{RA}$ | T, U, DT2m, W, NO$_2$, O$_3^{RA}$ |
| 4 | U, U10m, MSLP, SP, T2m, SH, W, CO, NO, NO$_2$, O$_3^{RA}$ | - |
| 5 | U, U10m, MSLP, SP, T2m, W, CO, NO, NO$_2$, O$_3^{RA}$ | T2m |
| 6 | U, V, U10m, V10m, MSLP, SP, T2m, SH, CO, NO | - |
| 7 | U, U10m, MSLP, SP, T2m, SH, W, CO, NO, NO$_2$ | U, T2m |

| | | |
|---|---|---|
| 8 | U, U10m, MSLP, SP, T2m, SH, W, CO, NO, $NO_2$, $O_3^{RA}$ | TCC, T2m |
| 9 | T, U, V, U10m, V10m, MSLP, SP, T2m, SH, W, CO, NO, $NO_2$ | - |
| 10 | T, U, U10m, MSLP, SP, T2m, W, NO, $O_3^{RA}$ | TCC, FCC, U, V, U10m, V10m, MSLP, SP, T2m, DT2m, SH, $SO_2$, NO, $NO_2$, $O_3^{RA}$ |
| 11 | U, U10m, MSLP, SP, T2m, W, CO, NO, $NO_2$, $O_3^{RA}$ | TCC, FCC, U, W |
| 12 | T, U, U10m, MSLP, SP, T2m, W, NO, $NO_2$, $O_3^{RA}$ | TCC |
| 13 | MSLP, SP, T2m, SH, W, NO, $O_3^{RA}$ | - |
| 14 | T, U, U10m, MSLP, SP, T2m, SH, W, NO, $O_3^{RA}$ | TCC, FCC, T2m, DT2m, $SO_2$, $O_3^{RA}$ |
| 15 | U, U10m, MSLP, SP, T2m, W, NO, $O_3^{RA}$ | TCC, U |
| 16 | MSLP, SP, T2m, W, NO, $O_3^{RA}$ | T, U, SP, W, $O_3^{RA}$ |
| 17 | T2m, $O_3^{RA}$ | - |
| 18 | T2m, SH, W, NO, $O_3^{RA}$ | TCC, FCC, DT2m, W, $O_3^{RA}$ |
| 19 | T, V10m, T2m, W, NO, $O_3^{RA}$ | TCC, FCC, V10m |
| 20 | T, V, V10m, SP, T2m, SH, $NO_2$, $O_3^{RA}$ | TCC, SP, T2m, W, $NO_2$, $O_3^{RA}$ |
| 21 | T, V10m, T2m,W, CO, $O_3^{RA}$ | CC, U, SP, SH, W, $SO_2$, $O_3^{RA}$ |
| 22 | T, V10m, MSLP, SP, T2m, W, $O_3^{RA}$ | TCC, FCC, U, V10m, MSLP, SP, SH, $O_3^{RA}$ |
| 23 | T, T2m, DT2m, W, $O_3^{RA}$ | SH |
| 24 | T2m, $O_3^{RA}$ | - |
| 25 | - | DT2m, CO |
| 26 | T, V, V10m, MSLP, SP, T2m, CO, $O_3^{RA}$ | - |
| 27 | T, V, U10m, V10m, MSLP, SP, T2m, CO | - |

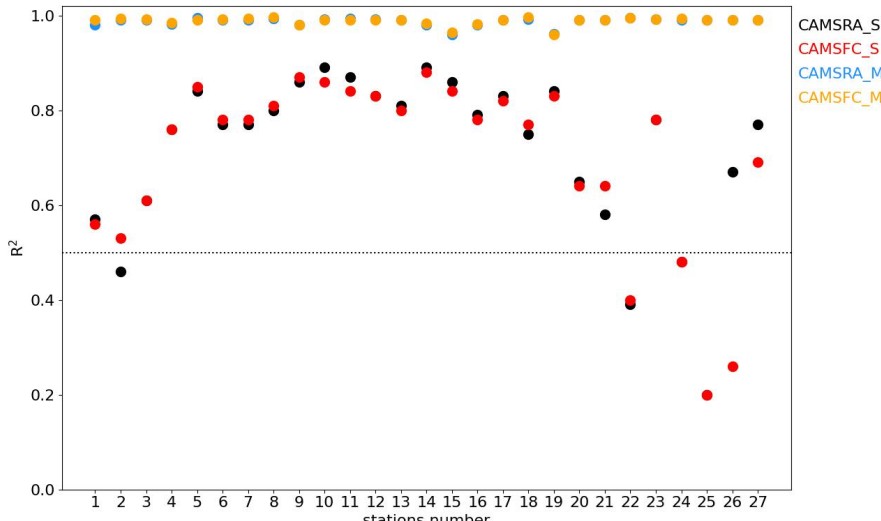


**Figure 5. The R² of the LSTM model for both S and M components of O₃. In this figure, CAMSRA_S and CAMSFC_S refer to the S components of CAMSRA and CAMSFC, respectively. Likewise, CAMSRA_M and CAMSFC_M refer to the M component of CAMSRA and CAMSFC, respectively.**


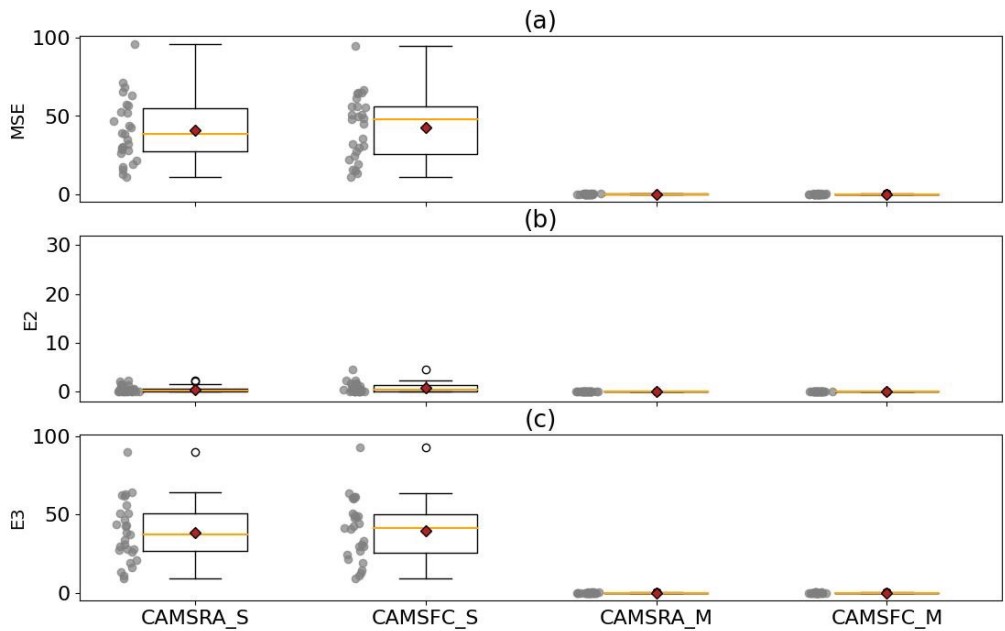

**Figure 6. The (a) MSE, (b) E2, and (c) E3 of the downscaled $O_3^{RA}$ and $O_3^{FC}$ with LSTM for both S and M components. In this figure, CAMSRA_S and CAMSFC_S refer to the S components of CAMSRA and CAMSFC, respectively. Likewise, CAMSRA_M and CAMSFC_M refer to the M component of CAMSRA and CAMSFC, respectively.**

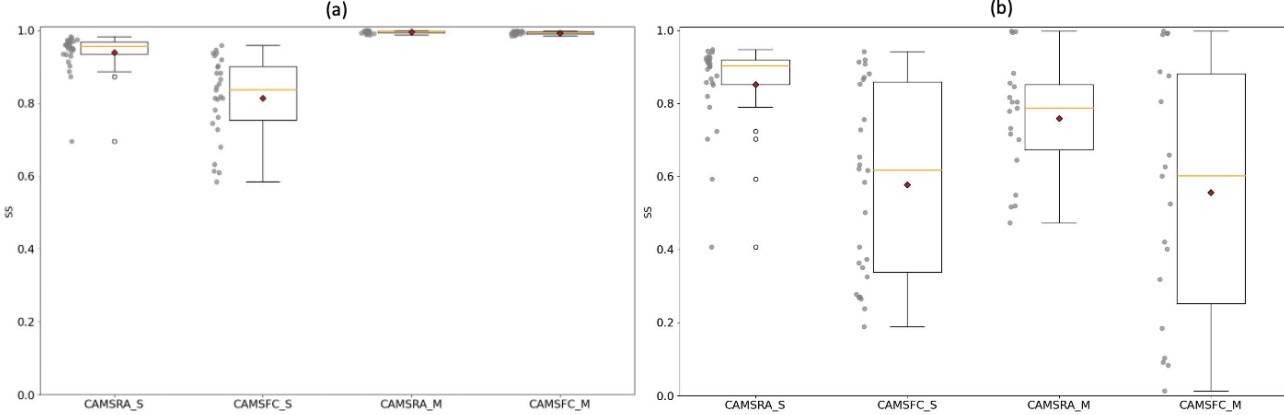

**Figure 7. The SS of the downscaled $O_3^{RA}$ and $O_3^{FC}$ with (a) LSTM and (b) LSTM$^{no\_lag}$**

**Appendix A**

**Table A1. The stations' names and their geographical locations**

| Number | Name | Latitude | Longitude | Number | Name | Latitude | Longitude |
|---|---|---|---|---|---|---|---|
| 1 | Abresan (Tabriz) | 38.066 | 46.326 | 15 | Shad abad (Tehran) | 35.67 | 51.297 |
| 2 | Namaz square (Tabriz) | 38.079 | 46.289 | 16 | Mahallati (Tehran) | 35.661 | 51.466 |
| 3 | Azarbayejan square (Tabriz) | 38.112 | 46.276 | 17 | District 19 (Tehran) | 35.635 | 51.362 |
| 4 | Aqdasiyeh (Tehran) | 35.795 | 51.484 | 18 | Masoudieh (Tehran) | 35.63 | 51.499 |
| 5 | Sadr (Tehran) | 35.778 | 51.429 | 19 | Ray (Tehran) | 35.604 | 51.426 |
| 6 | District 2 (Tehran) | 35.777 | 51.368 | 20 | Hamedan (Hamedan) | 34.8 | 48.5 |
| 7 | Punak (Tehran) | 35.762 | 51.332 | 21 | Birjand (Khorasan Jonoubi) | 32.87 | 59.21 |
| 8 | Geophysics (Tehran) | 35.74 | 51.385 | 22 | Yazd manabe tabiei (Yazd) | 31.93 | 54.37 |

| | | | | | | | | |
|---|---|---|---|---|---|---|---|---|
| 9 | Setad bohran (Tehran) | 35.727 | 51.431 | | 23 | Rasht (Gilan) | 37.29 | 49.61 |
| 10 | Tarbiat Modares (Tehran) | 35.717 | 51.386 | | 24 | Zanjan ark (Zanjan) | 36.67 | 48.48 |
| 11 | Sharif university (Tehran) | 35.702 | 51.351 | | 25 | Mirzaye shirazi (Markazi) | 34.09 | 49.7 |
| 12 | District 21 (Tehran) | 35.698 | 51.243 | | 26 | Kazeroon gate (Shiraz) | 29.61 | 52.53 |
| 13 | Piroozi (Tehran) | 35.696 | 51.494 | | 27 | Imam Hossein square (Shiraz) | 29.62 | 52.54 |
| 14 | Fath square | 35.679 | 51.337 | | | | | |


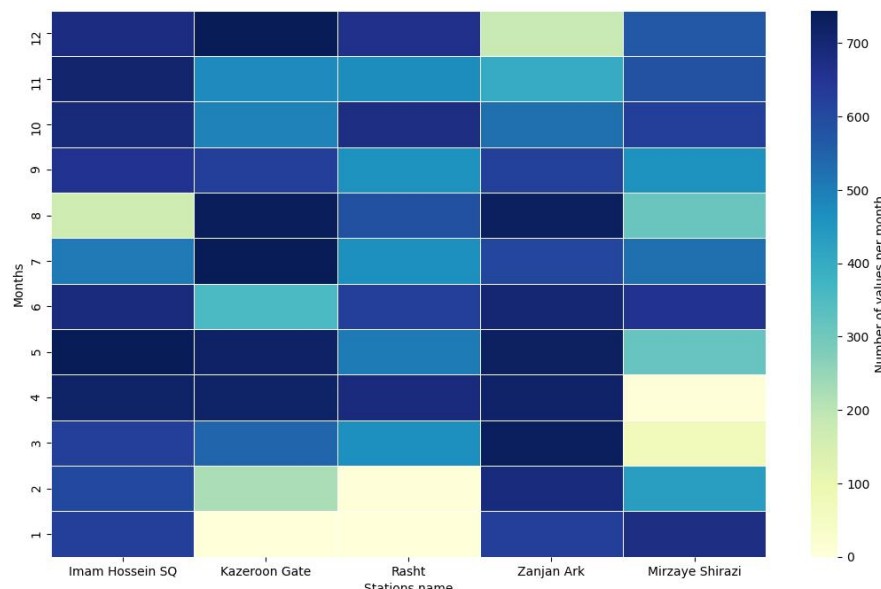

**Figure A1. Data coverage (per month) of the hourly surface-based measured ozone at five air quality monitoring stations**

**Table A2. The hyperparameter settings of the LSTM model**

| hyperparameter | values |
|---|---|
| Train portion | 65 % |
| Test portion | 35 % |
| Epoch | 1…30 |
| Batch size | [12, 24, 48, 72, 96, 120] |

| | |
|---|---|
| Optimizer | ADAM |
| Units (hidden layer) | 2…10 |
| Dropout rate | 0.001 |
| Learning rate | 0.001…0.1 |
| Loss function | MSE |


**Table A3. A list of the meteorological variables that were extracted from CAMS data products. $\oplus$ and $\ominus$ present available and unavailable variables, respectively.**

| Meteorological variable (symbol) | Units | Definition | CAMSRA | CAMSFC |
|---|---|---|---|---|
| T | °K | Temperature | $\oplus$ | $\oplus$ |
| T2m | °K | 2 meter temperature | $\oplus$ | $\oplus$ |
| DT2m | °K | 2 metre dewpoint temperature | $\oplus$ | $\oplus$ |
| SH | kg kg$^{-1}$ | Specific humidity | $\oplus$ | $\oplus$ |
| U | m s$^{-1}$ | U component of wind | $\oplus$ | $\oplus$ |
| V | m s$^{-1}$ | V component of wind | $\oplus$ | $\oplus$ |
| U10m | m s$^{-1}$ | 10 meter U wind component | $\oplus$ | $\oplus$ |
| V10m | m s$^{-1}$ | 10 meter V wind component | $\oplus$ | $\oplus$ |
| W | Pa s$^{-1}$ | Vertical velocity | $\oplus$ | $\oplus$ |
| BLH | m | Boundary layer height | $\ominus$ | $\oplus$ |
| SP | Pa | Surface pressure | $\oplus$ | $\oplus$ |
| MSLP | Pa | Mean sea level pressure | $\oplus$ | $\ominus$ |
| TCC | % | Total cloud cover | $\oplus$ | $\oplus$ |
| FCC | % | Fraction of cloud cover | $\oplus$ | $\oplus$ |
| UV | J m$^{-2}$ | Downward UV radiation at the surface | $\ominus$ | $\oplus$ |
| SD | s | Sunshine duration | $\ominus$ | $\oplus$ |


**Table A4. As Table 2, but for CAMSFC datasets.**

| Stations' number | S | M |
|---|---|---|
| 1 | DT2m, BLH, U10m, W, T, SH, $NO_2$, $NO_x$, CO | U, $O_3^{FC}$ |
| 2 | BLH, W, T, U | U, SP, $O_3^{FC}$ |
| 3 | BLH, T2m, U10m, W, T, U, $O_3^{FC}$ | T, U, $O_3^{FC}$ |
| 4 | BLH, V10m, V, $O_3^{FC}$ | - |
| 5 | V10m, $O_3^{FC}$ | $O_3^{FC}$ |
| 6 | BLH, U10m, V10m, T, U, V, SP, $O_3^{FC}$ | SD |
| 7 | BLH, T2m, V10m, W, T, V, $SO_2$, $O_3^{FC}$ | $O_3^{FC}$ |
| 8 | BLH, V10m, W, T, V, $O_3^{FC}$ | SD, U, $O_3^{FC}$ |
| 9 | BLH, T2m, V10m, W, T, V, NO, $SO_2$, CO, $O_3^{FC}$ | - |
| 10 | V10m, $O_3^{FC}$ | BLH, T, $O_3^{FC}$ |
| 11 | BLH, V10m, W, V, $O_3^{FC}$ | SD, $O_3^{FC}$ |
| 12 | BLH, V10m, $O_3^{FC}$ | $O_3^{FC}$ |
| 13 | BLH, V10m, T, V, $O_3^{FC}$ | $O_3^{FC}$ |
| 14 | BLH, V10m, V, SP, $NO_x$, $SO_2$, CO, $O_3^{FC}$ | DT2m, V10m, $O_3^{FC}$ |
| 15 | T2m, V10m, $O_3^{FC}$ | SD, BLH, $O_3^{FC}$ |
| 16 | BLH, U10m, V10m, T, V, $O_3^{FC}$ | DT2m, BLH, T2m, U10m, V10m, W, T, U, V, SH, SP, NO, $SO_2$, CO, $O_3^{FC}$ |
| 17 | T2m, $O_3^{FC}$ | BLH, V10m, $O_3^{FC}$ |
| 18 | BLH, V10m, W, T, V, NO, $O_3^{FC}$ | SD, T2m, U10m, V10mWu, V, $NO_2$, NO, $NO_x$, $SO_2$, CO, $O_3^{FC}$ |
| 19 | BLH, V10m, $O_3^{FC}$ | TCC, $O_3^{FC}$ |
| 20 | DT2m, BLH, T2m, U10m, T, U, SH, $O_3^{FC}$ | TCC, BLH, W, Q, SP, $O_3^{FC}$ |
| 21 | BLH, V, $SO_2$ | BLH, T, SP, $SO_2$, CO, $O_3^{FC}$ |
| 22 | DT2m, SD, BLH, T2m, U10m, V10m, W, T, U, V, SH, SP, $NO_2$, CO, $O_3^{FC}$ | BLH, U10m, SH, SP, $SO_2$ |
| 23 | U10m, $O_3^{FC}$ | SH |
| 24 | T2m, $O_3^{FC}$ | - |
| 25 | BLH | DT2m, BLH, $O_3^{FC}$ |

| 26 | BLH | - |
| 27 | SP | - |


**Table A5. The optimum units, dropout, learning rate and batch size to perform the LSTM model. Here, the T and F refer to True and False.**


| models | CAMSRA | | CAMSFC | |
|---|---|---|---|---|
| Stations' number | S | M | S | M |
| 1 | 10, T, 0.04, 24 | 10, T, 0.04, 24 | 2, T, 0.09, 48 | 4, F, 0.04, 48 |
| 2 | 4, F, 0.04, 48 | 4, F, 0.04, 48 | 10, T, 0.04, 24 | 10, T, 0.04, 24 |
| 3 | 10, T, 0.04, 24 | 4, F, 0.04, 48 | 2, T, 0.09, 48 | 4, F, 0.04, 48 |
| 4 | 4, F, 0.04, 48 | 4, F, 0.04, 48 | 10, T, 0.04, 24 | 4, F, 0.04, 48 |
| 5 | 10, T, 0.04, 24 | 4, F, 0.04, 48 | 10, T, 0.04, 24 | 4, F, 0.04, 48 |
| 6 | 10, T, 0.04, 24 | 10, T, 0.04, 24 | 10, T, 0.04, 24 | 10, T, 0.04, 24 |
| 7 | 10, T, 0.04, 24 | 4, F, 0.04, 48 | 10, T, 0.04, 24 | 4, F, 0.04, 48 |
| 8 | 10, T, 0.04, 24 | 10, T, 0.04, 24 | 2, T, 0.09, 48 | 10, T, 0.04, 24 |
| 9 | 4, F, 0.04, 48 | 10, T, 0.04, 24 | 4, F, 0.04, 48 | 10, T, 0.04, 24 |
| 10 | 10, T, 0.04, 24 | 4, F, 0.04, 48 | 4, F, 0.04, 48 | 4, F, 0.04, 48 |
| 11 | 10, T, 0.04, 24 | 10, T, 0.04, 24 | 4, F, 0.04, 48 | 10, T, 0.04, 24 |
| 12 | 4, F, 0.04, 48 | 10, T, 0.04, 24 | 2, T, 0.09, 48 | 10, T, 0.04, 24 |
| 13 | 4, F, 0.04, 48 | 4, F, 0.04, 48 | 10, T, 0.04, 24 | 4, F, 0.04, 48 |
| 14 | 2, T, 0.09, 48 | 2, T, 0.09, 48 | 10, T, 0.04, 24 | 4, F, 0.04, 48 |
| 15 | 4, F, 0.04, 48 | 10, T, 0.04, 24 | 4, F, 0.04, 48 | 10, T, 0.04, 24 |
| 16 | 10, T, 0.04, 24 | 10, T, 0.04, 24 | 10, T, 0.04, 24 | 10, T, 0.04, 24 |
| 17 | 10, T, 0.04, 24 | 10, T, 0.04, 24 | 10, T, 0.04, 24 | 10, T, 0.04, 24 |
| 18 | 4, F, 0.04, 48 | 4, F, 0.04, 48 | 4, F, 0.04, 48 | 10, T, 0.04, 24 |
| 19 | 10, T, 0.04, 24 | 2, T, 0.09, 48 | 10, T, 0.04, 24 | 2, T, 0.09, 48 |

| 20 | 2, T, 0.09, 48 | 10, T, 0.04, 24 | 4, F, 0.04, 48 | 4, F, 0.04, 48 |
| 21 | 2, T, 0.09, 48 | 4, F, 0.04, 48 | 10, T, 0.04, 24 | 10, T, 0.04, 24 |
| 22 | 10, T, 0.04, 24 | 10, T, 0.04, 24 | 10, T, 0.04, 24 | 4, F, 0.04, 48 |
| 23 | 4, F, 0.04, 48 | 10, T, 0.04, 24 | 10, T, 0.04, 24 | 10, T, 0.04, 24 |
| 24 | 4, F, 0.04, 48 | 10, T, 0.04, 24 | 4, F, 0.04, 48 | 10, T, 0.04, 24 |
| 25 | 10, T, 0.04, 24 | 10, T, 0.04, 24 | 10, T, 0.04, 24 | 10, T, 0.04, 24 |
| 26 | 10, T, 0.04, 24 | 10, T, 0.04, 24 | 4, F, 0.04, 48 | 10, T, 0.04, 24 |
| 27 | 4, F, 0.04, 48 | 10, T, 0.04, 24 | 2, T, 0.09, 48 | 10, T, 0.04, 24 |


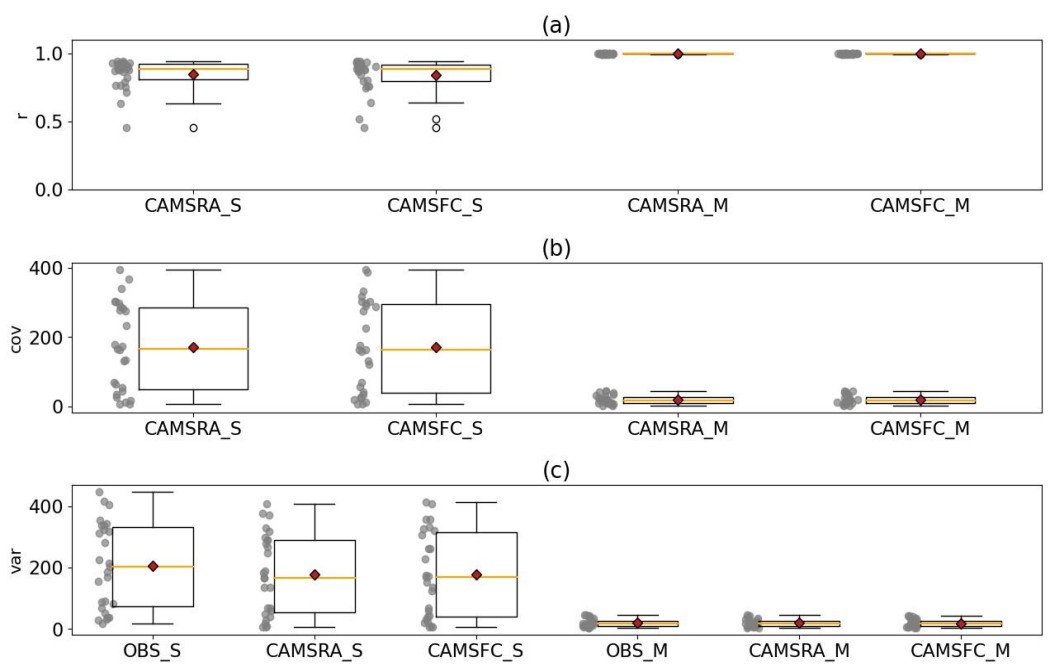

**Figure A2. The (a) correlation (r), (b) covariance (cov), and (c) variance (var) of the $O_3^{SD}$ with LSTM.**


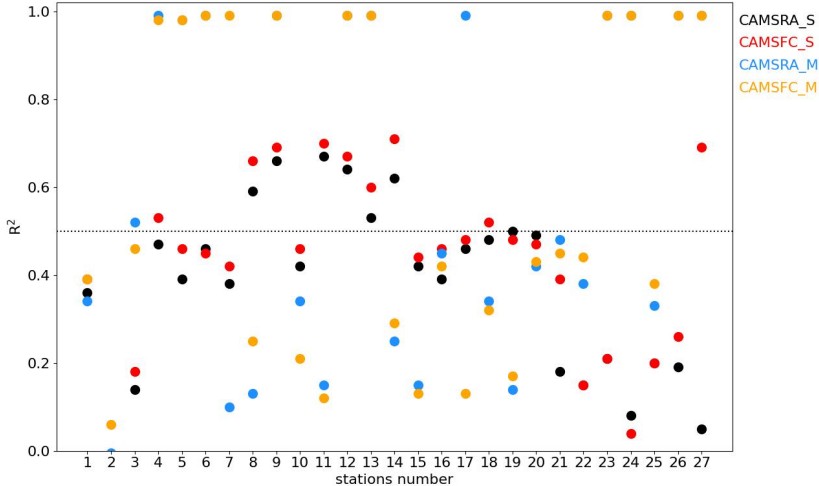

**Figure A3. As Fig. 5, but for the LSTM$^{no\_lag}$ model.**

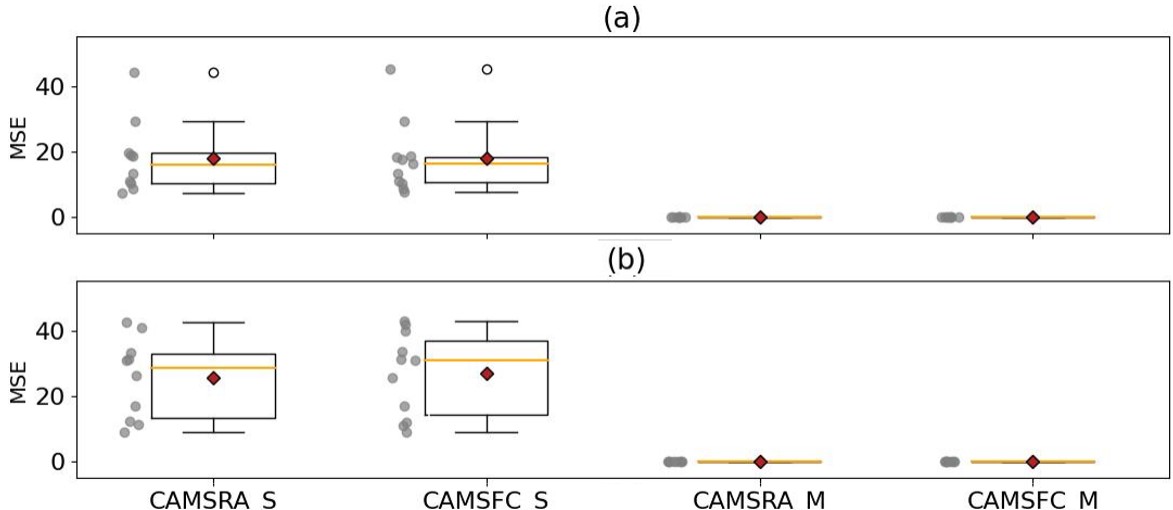

**Figure A4. The MSE of the $O_3^{SD}$ at the stations (excluding the stations over Tehran city) for (a) the cold {months = 1 to 3, and 10 to 12} and (b) the warm {months = 4 to 9} seasons, respectively.**

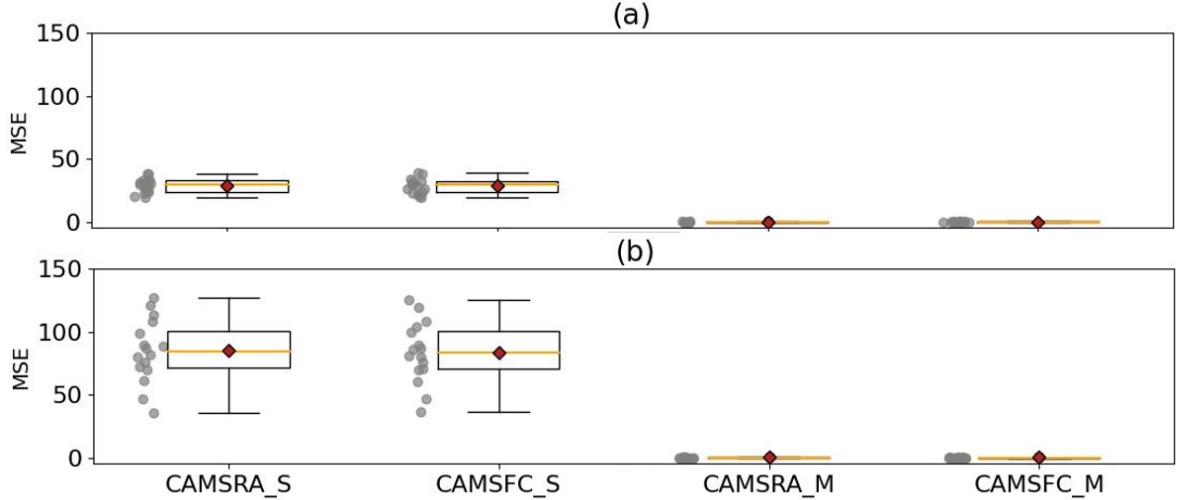

**Figure A5.** The MSE of the $O_3^{SD}$ at the stations over Tehran for (a) the cold {months = 1 to 3, and 10 to 12} and (b) the warm {months = 4 to 9} seasons.

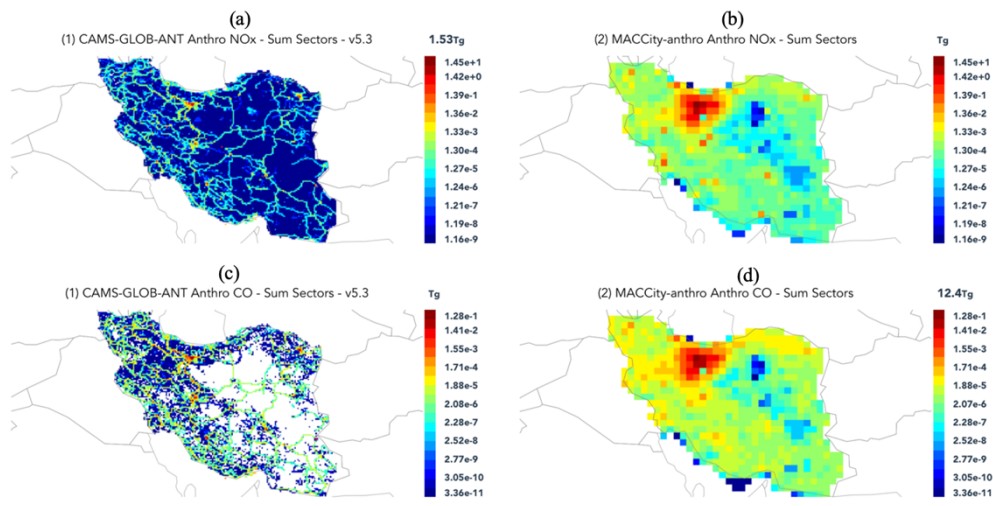

**Figure A6.** The annual average of surface emissions of the (a)-(b) $NO_x$ and (c)-(d) CO in the CAMS-GLOB-ANT and MACCity emission inventories.

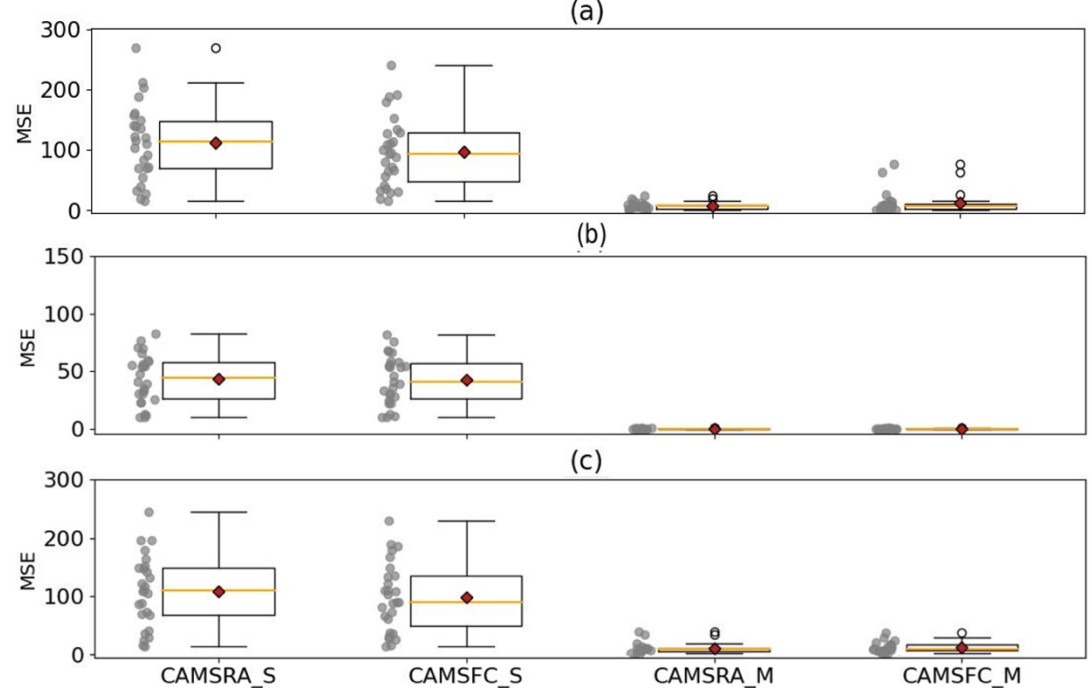

**Figure A7.** The MSE of the $O_3^{SD}$ by the (a) $LSTM^{no\_lag}$, (b) MLR, and (c) $MLR^{no\_lag}$ models


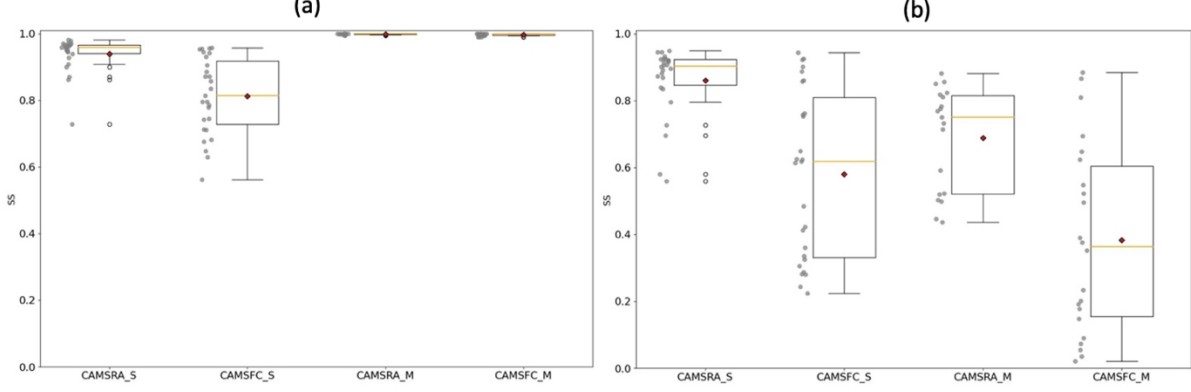

**Figure A8.** As Fig. 7 but for the downscaled data with (a) MLR and (b) $MLR^{no\_lag}$ models.


**Table A6.** The most important explanatory variables of the models at most of the stations.

|  | Meteorological variables | Chemical species |
|---|---|---|
| **CAMSRA_S** | T2m | NO, NO$_2$, $O_3^{RA}$ |
| **CAMSFC_S** | BLH, V10m | $O_3^{FC}$ |
| **CAMSRA_M** | TCC, U | $O_3^{RA}$ |
| **CAMSFC_M** | - | $O_3^{FC}$ |


**Table A7.** The results of the experiments (1) $MLR^{no\_lag(expr1)}$: the model was trained only using $O_3^{RA}$ and $O_3^{FC}$, (2) $MLR^{no\_lag(expr2)}$: the model was trained using the meteorological variables with high priority (listed in Table A6) at station 22 (Yazd). The *r* refers to the correlation coefficient between $O_3^{SD}$ and measured $O_3$.

| | $MLR^{no\_lag}$ | | $MLR^{no\_lag(expr1)}$ | | $MLR^{no\_lag(expr2)}$ | |
|---|---|---|---|---|---|---|
| | MSE | *r* | MSE | *r* | MSE | *r* |
| **CAMSRA_S** | 14.94 | 0.41 | 16.06 | 0.33 | 16.09 | 0.32 |
| **CAMSFC_S** | 14.69 | 0.43 | 16.30 | 0.31 | 16.01 | 0.33 |
| **CAMSRA_M** | 1.85 | 0.61 | 2.81 | 0.22 | 2.92 | 0.10 |
| **CAMSFC_M** | 1.77 | 0.63 | 2.90 | 0.12 | - | - |


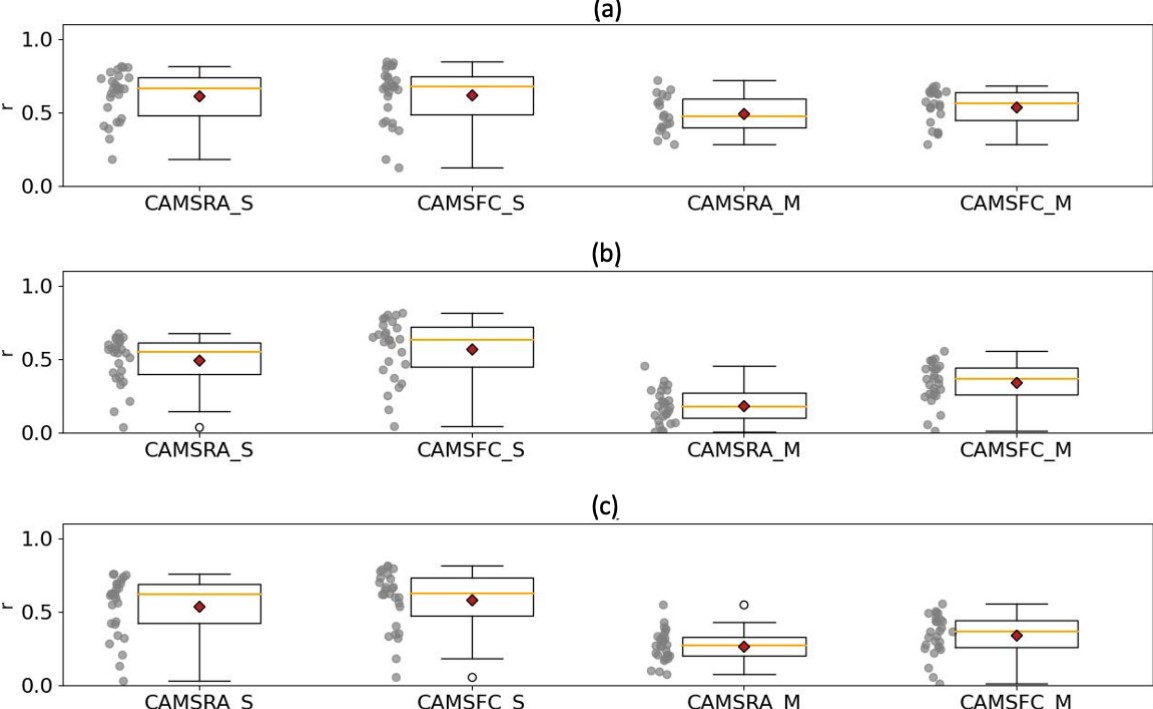

**Figure A9.** The correlation (*r*) between measured $O_3$ and $O_3^{SD}$ by the (a) $MLR^{no\_lag}$, (b) $MLR^{no\_lag\,(expr1)}$, and (c) $MLR^{no\_lag\,(expr2)}$ models.