# Peer review of "Assessment of surface ozone products from downscaled CAMS reanalysis and CAMS daily forecast using urban air quality monitoring stations in Iran"

_Geoscientific Model Development, 2023_

## Author Comment (AC2)

**Authors' response to reviewer #1**

Please note that in this document, colors' codes are as the referee's comments in black, the authors' responses in blue. Authors' changes in the manuscripts are shown in blue bold. The tables and figures are referred based on their numbers in the revised manuscript with markup (RM).

The authors present a thorough investigation of the variability in surface ozone of two related CAMS products compared to a comprehensive set of ozone observations distributed over Iran. To account for the fact that the global model simulations are not optimized for these conditions the authors have developed a downscaling approach based on a so-called LSTM neural network method, with, apart from modeled ozone, also assimilated meteorological quantities, as well as lagged $O_3$ observations. They show the benefit of the LSTM method compared to using the raw CAMS products for providing $O_3$. Also particularly the importance of the lagged $O_3$ observations was quantified.

The authors highly acknowledge the referee for spending time and providing valuable and punctilious comments.

I consider this manuscript well suited for publication in GMD, considering the comprehensive analyses presented here, including the development of the LSTM method, and the analysis of the different model versions to represent the ozone variability at various temporal scales and different stations and regions over Iran, which is also of wider interest. My only main hesitation concerns the difficulty to follow exactly what approach the authors have taken in their methodology. Any further revisions that help to clarify (and improve readability) their methods is still welcome. Also some further discussion on the implications of your study, e.g., for the use of coarse-scale global model data such as CAMS for policy applications (?) or possibly the scope of the methods developed here for wider application (?) should be better highlighted, to better place this work in a wider context. Part of the scope of this work is indeed mentioned on line 56-60 of the introduction, but there is no explicit answer to these interesting research questions in the abstract or the conclusions - only indirectly.

- In the RM, the methodology was improved from two perspectives, i.e., grammatically and context (adding more details), as follows:

L174-176: **This section has been divided into three sections. Sect. 3.1 details the theory of decompositions and the method used in this study. Sect. 3.2 describes the procedure for neural network modeling and the pre-processing of its input. Sect. 3.3 defines the metrics (indicators) that have been used to assess the CAMS performance and error sources.**

L199-204: **As expected from Eq. (1), KZ (5, 5) filters all periods less than 11.2 time steps. This corresponds to 33.54 hours or 1.4 days, as the data are recorded in an interval of 3-hours. The same holds for KZ (35, 5), which filters all periods less than 9.8 days. Hence, the S refers to the short scale fluctuations which is done in less than 1.4 days. Similarly, M refers to the synoptic scales events with a time scales ranging from 1.4 to 9.8 days. The variations with the time scales of more than 9.8 days are represented in L term.**

L217-237: **A range of control values for several hyperparameters (Table A2) were tested by multiple trial-and-errors. The most effective hyperparameters (Table A5) were selected using the Random Search optimization method. To prepare the LSTM inputs, several meteorological**

**variables (Table A3) were obtained from the CAMSRA and CAMSFC datasets. To prevent overfitting of the model, a cross-validation Lasso regression was performed to identify the potential predictors at each station. The lagged $O_3$ (from OBS) was also considered as the model inputs, since the concentration of $O_3$ is not only affected by meteorological factors but also by the influence of the $O_3$ levels in the past.**

L259-264: **That could arise from overfitting associated with complex chemical processes in the model or imbalance among coupled components. The E3 represents an unexplained error, reflecting the lack of observed variability in the modelled data. That refers to the variabilities which are not captured by the models, even though those variabilities exist in the observations. The E3 can arise from random and non-representative errors caused by sub-scales and non-resolvable processes in the observations, or from a deficiency of the model in capturing meso-scale phenomena.**

- Our methodology can be used for the validation of the other chemical species (simulated by global models). Besides, it can be used for the predictions of chemical species. Accordingly, the following text was added to the end of abstract as:

L20-22: **This study demonstrates that coarse-scale global model data such as CAMS needs to be downscaled for regulatory purposes or policy applications at local scales. Our method can be useful not only for the evaluation but also for the prediction of other chemical species, such as aerosols.**

- Yes, that was a missing point. An explicit answer to the questions were added in the abstract:

L15-17: **Results show the benefit of the LSTM method compared to using the raw CAMS products for providing $O_3$ over Iran. It is found that lagged $O_3$ observation has a larger contribution than other predictors in improving the LSTM.**

- The implication and an explicit answer to the questions were added in the conclusions as:

L616-620: **To date, most of the studies of ozone and other pollutants in Iran rely on reanalysis products, without using decompositions or downscaling procedures. Our findings show that the CAMSRA and CAMSFC datasets have some deficiencies in simulating ozone, in particular over the cities with high emissions of ozone precursors. Downscaling improves these products and makes them suitable for the study of ozone in major metropolitan areas. The method used in this study is not only applicable for the evaluation of the global models but also for prediction purposes.**

**More specific comments:**

- L16: "correspondence precision" - not clear what this is - suggest to use another wording here.

Right, the word "association" could be a better choice. That was modified as

L17: **more associations**

- end of abstract (and end of conclusions): I expect a sentence that briefly describes the implications of your study. Same comment holds for the (end of) the conclusions.

  Right. Those were missing points which were added at the end of abstract and the end of conclusions as:

  L20-22: **This study demonstrates that coarse-scale global model data such as CAMS needs to be downscaled for regulatory purposes or policy applications at local scales. Our method can be useful not only for the evaluation but also for the prediction of other chemical species, such as aerosols.**

  L616-620: **To date, most of the studies of ozone and other pollutants in Iran rely on reanalysis products, without using decompositions or downscaling procedures. Our findings show that the CAMSRA and CAMSFC datasets have some deficiencies in simulating ozone, in particular over the cities with high emissions of ozone precursors. Downscaling improves these products and makes them suitable for the study of ozone in major metropolitan areas. The method used in this study is not only applicable for the evaluation of the global models but also for prediction purposes.**

- L41: suggest to change this sentence to: "In recent years, the Copernicus Atmosphere Monitoring Service (CAMS) has been mainly developed to assimilate observations of chemical composition to provide analyses of tropospheric ozone and aerosol concentrations,… "

  That was modified as

  L61-62: **In recent years, the Copernicus Atmosphere Monitoring Service (CAMS) has been mainly developed to assimilate observations of chemical compositions to provide analyses of tropospheric ozone and aerosol concentrations, …**

- L43: "and a control run (no assimilation)" -> "and a control run (without assimilation of atmospheric composition)"

  That was modified as

  L64-65: **a control run (without assimilation of atmospheric composition).**

- L75: "…using a four-dimensional variational (4D-Var) scheme as…

That was modified as

L102: **using a four-dimensional variational (4D-Var) scheme as …**

- L80: "MERAA"-> "MERRA"

A typo; that was corrected.

L106: **MERRA**

- L84: "It is noteworthy that newer versions of data have been frequently adopted in CAMS." : it is unclear what the authors want to convey in this sentence. Is it that different satellite data have been used in the Reanalysis product, or that different CAMS reanalysis products exist, with CAMSRA the latest and most comprehensive, to date?

The adoptions refers to the CAMS upgrades such as improving horizontal resolutions, vertical levels, newer version of the satellite retrievals. CAMS uses various satellite observations, covering different time periods (Table 2 in Innes et al., 2019). So the text was modified as:

L111-112: **CAMS carries several upgrades, such as improving horizontal resolutions, vertical levels, and the newer version of the satellite retrievals. CAMS uses various satellite observations, covering different time periods.**

- L91: "Compared to CAMSRA, in CAMSFC only the initial conditions of each forecast are obtained from reanalysis datasets, i.e.,.." consider change to "Compared to CAMSRA, in CAMSFC the initial conditions of each forecast are obtained from analyses of atmospheric composition in near-real time, i.e.,…"

That was modified as

L122-123: **Compared to CAMSRA, in CAMSFC the initial conditions of each forecast are obtained from analysis of atmospheric composition in near-real time,**

- L98: "Biomass burning injects from GFAS" -> "biomass burning emissions are based on GFAS".

That was modified as

L130: **Biomass burning emissions are based on GFAS.**

- L102: "from 9 July 2019 onwards**,…**"

Right. That was modified.

L133: **From 9 July 2019 onwards,**

- L149-150: "KZ(35,5)"  - I understand that 35 here refers to 'm', the window size. But can the authors please explain why they choose the value of 35 here? (and a value of 5 in the definition of S in eqn. 2) Does this correspond to a filtering time scale of 35 x 3hr = approx. 105 hr, i.e. 4 days?

Yes, the values are corresponds to the filtering time scales. Based on Eq. (1):

$35 \times \sqrt{5} = 78.3$ (time steps)

As the data are 3-hourly so:

$78.3 \times 3 = 234.8$ (hours) $\approx 9.8$ (days)

This point was clarified in RM as:

L199-204: **As expected from Eq. (1), KZ (5, 5) filters all periods less than 11.2 time steps. This corresponds to 33.54 hours or 1.4 days, as the data are recorded in an interval of 3-hours. The same holds for KZ (35, 5), which filters all periods less than 9.8 days. Hence, the S refers to the short scale fluctuations which is done in less than 1.4 days. Similarly, M refers to the synoptic scales events with a time scales ranging from 1.4 to 9.8 days. The variations with the time scales of more than 9.8 days are represented in L term.**

- L189-191: As a modeler on initial reading I find this split in definition between 'explained' and 'unexplained' error a bit artificial. Different to what is suggested, I would also not have a direct understanding of the cause of 'explained error'. After reading the manuscript, I think I better understand the arguments of calling errors either 'explained' or 'unexplained', but it might help to allude to that.

The explained error refers to errors, which arise from the model, i.e., $\sigma_m - r\,\sigma_o$, in Eq. (5). For instance, the model shows some variabilities, which are unseen in the observed data. On the other hand, the unexplained error refers to the variabilities which are not captured by the model, even though those variabilities exist in the observations $\sigma_o^2(1 - r^2)$ in Eq. (5). This point was explained in the RM as:

L259-264: **That could arise from overfitting associated with complex chemical processes in the model or imbalance among coupled components. The E3 represents an unexplained error, reflecting the lack of observed variability in the modelled data. That refers to the variabilities which are not captured by the models, even though those variabilities exist in the observations. The E3 can arise from random and non-representative errors caused by sub-scales and non-resolvable processes in the observations, or from a deficiency of the model in capturing meso-scale phenomena.**

- L235: "for most of the station**s...**"

  Right. That was modified.

  L322: **for most of the stations**

- L246 and L248: The authors refer here to 'opochs'. please provide an explanation what an 'epoch' is exactly, in this context. I missed that.

  LSTM model requires a specific configuration and tuning to work effectively with the datasets. A range of control values was tested by multiple trial-error evaluations using the Scikit function GridSearchCV. In our experiments, we tune one of the hyperparameters, i.e., epoch number from 1 to 30. This value is a hyperparameters for the learning algorithm, e.g., parameters for the learning process, not internal model parameters found by the learning process. The number of epoch is traditionally large, often hundreds or thousands, allowing the learning algorithm to run until the error from the model (loss function) has been sufficiently minimized. There are no given rules to set this parameter. One epoch leads to underfitting of the curve. As the number of epochs increases, more number of times the weight are updated in the neural network and the curve goes from underfitting to optimal curve.

[Figure]

  In the RM, we applied an "Early Stopping" option, which allows to specify an arbitrary large number of training epochs and stop training once the model performance stops improving on a hold out validation dataset.

  L340-348: **We tuned hyperparameters, which allow the learning algorithm to run until the error from the model, i.e., the loss function, has been sufficiently minimized. As there**

**are no given values to set these numbers, the optimum values were obtained by multiple trial-and-error tests (see Table A5).**

- L250-251: "That might reflect that the more predictors, the better the model would not be." : as also reflected in the conclusions, I find this an important finding indeed. I'd suggest to stress this a bit better, also by re-formulating this sentence a little - now it reads a little clumsy. This finding may be worth a bit more statistical analysis, i.e. do the authors have any quantitative metric arising from the method which provides insight as to how much each of the individual parameters contributes to the quality of the end product? It might be a useful exercise to exclude some of the (physically) less obvious parameters from the list of fitting parameters, such as U, V, W, MSLP (?). Here an analysis of station 22 (Yazd), which performs relatively poor, while it uses an excessive list of input data, suggests indeed the limitations of this work. Can the authors comment?

That is indeed a very good point and suggestion. To decide on the importance of the variables we used LassoCV estimator. The variables with the highest absolute Lasso coefficient (importance weight) are considered the most important. For instance, Fig. 4 shows that the T2m is the most explanatory meteorological variable and $NO$, $NO_2$, and $O_3^{RA}$ are the main chemical variables for CAMSRA_S at most of the stations. The higher the weight value, the more the influence the variable has and hence more important. Table A6 lists the more influential variables on ozone variability at most of the stations.

To assess the sensitivity of the model to the less obvious predicators, we designed two experiments. In first experiment ($MLR^{no\_lag(expr1)}$), the model was trained only using $O_3^{RA}$ and $O_3^{FC}$, while in second experiment ($MLR^{no\_lag(expr2)}$), the model was trained using the meteorological variables with high priority (as listed in Table A6). Both experiment were preform using $MLR^{no\_lag}$.

Our analysis shows a low value of $R^2$ for the S component at station Yazd (22), but that is relatively significant for the M term. Besides, the MSE at this station is very low (so good model performance). That could be associated with the station locations, which are less populated and less affected by local anthropogenic emissions sources (and easier to model). That is not related to the excessive list of input data. As this experiment shows by excluding most of the parameters, the MSE at this stations changes from 14.94 to 16.06 (does not change that much).

L568-575: **Two experiments were designed to assess the sensitivity of the model to less obvious predictors. In the first experiment, i.e., $MLR^{no\_lag(expr1)}$, the model was trained only using $O_3^{RA}$ and $O_3^{FC}$. In the second experiment, i.e., $MLR^{no\_lag(expr2)}$, the model was trained using the most influential meteorological variables (see Table A6). For the sake of simplicity (and being less expensive), both experiments were performed using the $MLR^{no\_lag}$ model. Table A7 lists the results of these experiments for station 22 (Yazd). As can be seen, the MSE of $MLR^{no\_lag(expr1)}$ and $MLR^{no\_lag(expr2)}$ are larger than that of $MLR^{no\_lag}$. That shows that part of the $O_3$ variability is explained by meteorology and**

partly by the chemistry ($O_3{}^{RA}$ or $O_3{}^{FC}$). Separating these two factors causes a decline of $r$ (see Fig. A9).

[Figure]

**Figure 4. Cross-validation Lasso regression to identify the potential predictors for ozone modeling. The higher absolute Lasso coefficient, the most important would be the variable.**

**Table A6. The most important explanatory variables of the models at most of the stations.**

|  | **Chemical species** | **Meteorological variables** |
|---|---|---|
| **CAMSRA_S** | NO, NO$_2$, O$_3{}^{RA}$ | T2m |
| **CAMSFC_S** | O$_3{}^{FC}$ | BLH, V10m |
| **CAMSRA_M** | O$_3{}^{RA}$ | TCC, U |
| **CAMSFC_M** | O$_3{}^{FC}$ | - |

**Table A7. The results of the experiments (1) MLR$^{no\_lag(expr1)}$: the model was trained only using O$_3{}^{RA}$ and O$_3{}^{FC}$, (2) MLR$^{no\_lag(expr2)}$: the model was trained using the meteorological variables with high priority (listed in Table A6) at station 22 (Yazd). The $r$ refers to the correlation coefficient between O$_3{}^{SD}$ and measured O$_3$.**

|  | **MLR$^{no\_lag}$** | | **MLR$^{no\_lag(expr1)}$** | | **MLR$^{no\_lag(expr2)}$** | |
|---|---|---|---|---|---|---|
|  | MSE | $r$ | MSE | $r$ | MSE | $r$ |
| CAMSRA_S | 14.94 | 0.41 | 16.06 | 0.33 | 16.09 | 0.32 |
| CAMSFC_S | 14.69 | 0.43 | 16.30 | 0.31 | 16.01 | 0.33 |
| CAMSRA_M | 1.85 | 0.61 | 2.81 | 0.22 | 2.92 | 0.10 |
| CAMSFC_M | 1.77 | 0.63 | 2.90 | 0.12 | - | - |

[Figure]

**Figure A9. The correlation ($r$) between measured $O_3$ and $O_3^{SD}$ by the (a) MLR$^{no\_lag}$, (b) MLR$^{no\_lag}$ (expr1), and (c) MLR$^{no\_lag}$ (expr2) models.**

▪ L282: "lagged $O_3$" please specify here (again) that this refers to lagged $O_3$ from actual observations, to help the reader understand.

Sure, that was applied as

L429-430: **In order to examine the effect of the CAMS products and lagged $O_3$ (from actual observations) on the LSTM model, we exclude the measured lagged ozone from the predictors of the LSTM model.**

▪ L301: typo 'products'

That was corrected. This typo was also fixed in L9 and Table A3.

▪ L328: suggest to drop the sentence "These values…" - no need?

That was modified.

L507: **respectively, and increase to …**

- L364:"peroxides"->"proxies"

  Its sentences was deleted.

- L377-380: I'd expect here a comment on the implication of these findings, e.g. the importance of observed (lagged) $O_3$ as predictor (?) and/or the potential use cases of the methods as the authors have developed.

  Very good suggestion. So, the changes applied in the RM are as follow:

  L605: **That shows the importance of the observed (lagged) $O_3$ as a predictor in the LSTM.**

- Table 1: "single level" -> "surface level"

  Right. That was modified.

  Table 1: **surface level**

- Table A3: typo in units for UV

  That was corrected:

  Table A3: **$J\ m^{-2}$**

---

## Author Comment (AC3)

**Authors' response to reviewer #2**

Please note that in this document, colors' codes are as the referee's comments in black, the authors' responses in blue. Authors' changes in the manuscripts are shown in blue bold. The tables and figures are referred based on their numbers in the revised manuscript with markup (RM).

This study uses observational and CAMS reanalysis and forecast data to investigate $O_3$ variability and errors in CAMS global systems for several stations over Iran. To this end, the observed and CAMS $O_3$ time series are decomposed in three spectral components short, medium, and large, with the latter one not examined. Subsequently, an LSTM neural-network is applied to downscale the components and then investigate the CAMS performance and error sources.

Overall, I find this to be an interesting study, yet, with several points that need clarification and improvement, to suggest publication in GMD.

The authors highly acknowledge the referee for spending time and providing valuable and punctilious comments. An investigation of the L term could add more values to the article. But due to limitations in measured data (e.g., time series with long time periods over Iran), the assessment of this term was not possible. Most of the suggestions were applied in the RM; here are a few points to be mentioned.

**Main comments**

1. Title of the paper: The authors use surface-ozone data from both observations and CAMS (RA and FC). Is this correct? If yes, I don't understand the rationale behind the use of term "tropospheric ozone" in the title and in the abstract. Tropospheric ozone variability is governed by different processes and spatiotemporal scales compared to surface ozone, thus, I believe that the title is somehow misleading.

That is absolutely right. The term "Tropospheric" had been kept fixed since the early version of the paper, which was more concerned about ozone at upper levels. In the RM, the title was modified as:

L1: **Assessment of surface ozone products …**

2. I feel that more interpretation of the results is needed in the manuscript. For example, uncertainties in CAMS emissions inventories and deposition, and the fact that the two products use different emissions inventories should be discussed, especially for the S term.

Very important point. Emissions have large impact on both terms. Uncertainties in the spatial variations of the emissions inventories were discussed.

L361-363: **That could arise from the uncertainties in $O_3$ precursor emissions affecting modeled local photochemistry and likely S variability. The largest value of the MSE is associated with the $O_3^{SD}$ of the stations located in the city of Tehran. That can be associated with the uncertainties in CAMS emissions inventories, which may have larger impact in cites with high anthropogenic**

**emissions sources. The stations in the northern part of the city (e.g., stations 4, 5, 6, 7, 8, and 9) show a larger MSE than the stations in the southern part (e.g., stations 10, 11, 14, 15, 16, 17, and 19). That can be associated with the deficiency of the emissions inventories in capturing the local emissions changes within urban areas.**

**L370-373: That can be associated with the uncertainties in the spatial variations of the emissions inventories used in CAMS. Although the CAMS anthropogenic emission inventories account for emissions from different sectors, such as transportation, residential and energy sectors, as well as biogenic fluxes, they have a temporal and spatial allocation with a monthly spatial grid resolution of 0.1°x0.1°.**

**L380-418: The S variability results from the effect of daytime photochemical production, downward transport of $O_3$ rich from upper levels, combined with $O_3$ loss by depositions (in the surface layer).**

**L491-502: That could arise from the uncertainty of $O_3$ precursors in CAMS, as they are not adjusted by data assimilation systems. CAMS-GLOB-BIO (used in CAMSFC, see Table 1) provides monthly average of the global biogenic emissions, which are calculated using the MEGAN (used in CAMSRA, see Table 1), driven by ERA-Interim meteorological fields.**

**L511-516: The impact of these (meso scale) factors cannot be captured in a global emissions inventory with limited resolutions. That induces large model uncertainties, in particular for the S variability, which has large associations with pollutant species. Besides, for some periods the emissions are not available and so prescribed, which means they are either kept fixed since the last year available or extrapolated (projected) with a climatological trend. MACCity emission inventory has not been updated after 2010, and recent years are only based on projections of past trends. CAMS-GLOB-ANT provides the monthly average of the global emissions of 36 compounds over the period 2000-2019.**

3. The role of stratospheric ozone contribution to the surface ozone is only discussed once in the Discussion. Stratospheric ozone can affect surface ozone levels indirectly through vertical downward transport of ozone from the lower stratosphere and/or the upper troposphere in larger time scales (Zanis et al., 2014) or directly through intense stratospheric intrusions (rarer) (Akritidis et al., 2010, Chen et al., 2022). The broader Iran region is a well-known hot spot of stratosphere-to-troposphere transport that might affect day-to-day O3 variability in some cases. The CAMS reanalysis product includes a tracer for stratospheric ozone that might be useful tracer for stratospheric ozone. This is a suggestion that the authors might consider for the present or future studies. At least, the authors should consider for the discussion that by not including such a source (stratospheric ozone) of surface ozone is a potential source of error.

Good point. Yes, Iran is recognized as a hot spot for folding activities and rare deep folds, in particular in spring and summer. The fold frequency distribution over the eastern Mediterranean and the Middle East shows two distinct maxima, one of which is located over Iran. Several studies show that the subsidence of stratospheric ozone, under given weather systems over Iran. This point was further discussed in the RM.

Assessing of the stratospheric ozone tracer is a great idea, and could help in the investigation of the error sources. We consider this point for future study due to lack of time.

L520-531: **Stratospheric ozone can affect surface ozone levels indirectly through vertical downward transport of ozone from the lower stratosphere and/or the upper troposphere in larger time scales (Zanis et al., 2014) or directly through intense stratospheric intrusions (rarer) (Akritidis et al., 2010; Chen et al., 2022). Over Tehran, a major portion of O$_3$ during spring is transferred from the stratosphere (Aliakbari Bidokhti and Shariepour, 2007). A study by (Shariepour and Aliakbari Bidokhti, 2013) showed that several mid-latitude low pressure weather systems accompanied by tropopause folding affect northern Iran (Caspian Sea), and can cause downward transport of stratospheric ozone rich air towards the surface. During summer, the occurrence of tropopause folding and their intensity over the Eastern Mediterranean and the Middle East regions are majorly controlled by the Asian monsoon. Since the zone of upper level baroclinicity and fold occurrences spread northwestward over this region, it first reaches Iran (Tyrlis et al., 2014).**

L610-612: **In addition, uncertainties in stratospheric ozone and emission inventory might affect this error. Analysis of a tracer for stratospheric ozone can help to identify non-local ozone sources (a recommendation for future work).**

**Akritidis, D., Zanis, P., Pytharoulis, I., Mavrakis, A., and Karacostas, Th.: A deep stratospheric intrusion event down to the earth's surface of the megacity of Athens, Meteorol. Atmos. Phys., 109, 9–18, doi:10.1007/s00703-010-0096-6, 2010.**

**Aliakbari Bidokhti, A.A. and Shariepour, Z.: Analysis of surface ozone variability in the vicinity of synoptic (meteorology) station of Geophysics Institute (Tehran University) for the year 2002, J. Environ. Stud. 33 (42), 63–74, 20.1001.1.10258620.1386.33.42.7.7, 2007.**

**Chen, Z., Liu, J., Qie, X., Cheng, X., Shen, Y., Yang, M., Jiang, R., and Liu, X.: Transport of substantial stratospheric ozone to the surface by a dying typhoon and shallow convection, Atmos. Chem. Phys., 22, 8221–8240, https://doi.org/10.5194/acp-22-8221-2022, 2022.**

**Shariepour, Z. and Aliakbari Bidokhti, A.A.: Investigation of Surface Ozone over Tehran for 2008-2011, 39, 191–206, https://doi.org/10.22059/jesphys.2013.35607, 2013.**

4. I agree with the main comment raised by Reviewer 1. The manuscript should be more oriented to presenting the main objectives of the study and then the main implications. For sure, this will make it more reader friendly.

The main objective of the study is to show the capability of the reanalysis products in capturing the surface ozone variability over Iran. Many scientists in Iran use reanalysis datasets (instead of observational data) and provide advices for policy applications. This issue can extend to other chemical species. These points were considered in the RM as:

L15-16: **Results show the benefit of the LSTM method compared to using the raw CAMS products for providing O$_3$ over Iran.**

L20-22: **This study demonstrates that coarse-scale global model data such as CAMS needs to be downscaled for regulatory purposes or policy applications at local scales. Our method can be useful not only for the evaluation but also for the prediction of other chemical species, such as aerosols.**

L616-620: **To date, most of the studies of ozone and other pollutants in Iran rely on reanalysis products, without using decompositions or downscaling procedures. Our findings show that the CAMSRA and CAMSFC datasets have some deficiencies in simulating ozone, in particular over the cities with high emissions of ozone precursors. Downscaling improves these products and makes them suitable for the study of ozone in major metropolitan areas. The method used in this study is not only applicable for the evaluation of the global models but also for prediction purposes.**

**Comments**

- P1, L21: A recent review of air pollution impacts on health is worth cited here by Pozzer et al. (2023).

  Yes, that is right. It was added.

  L25: **Malley et al., 2015; Pozzer et al., 2023).**
  **Pozzer, A., Anenberg, S. C., Dey, S., Haines, A., Lelieveld, J., and Chowdhury, S.: Mortality attributable to ambient air pollution: A review of global estimates, GeoHealth, 7, https://doi.org/10.1029/2022GH000711, 2023.**

- P5, L135-136: "whereas long-term and seasonal variation is mainly related to solar radiation." What about long-range transport and stratosphere-to-troposphere transport (Monks et al., 2000).

  That was a missing point. It was added in the RM as

  L180-181: **whereas long-term and seasonal variations are mainly related to solar radiation, long-range transport and transport from the stratosphere to the troposphere (Monks, 2000).**

  **Monks, P. S.: A review of the observations and origins of the spring ozone maximum, Atmos. Environ., 34, 3545–3561, https://doi.org/10.1016/S1352-2310(00)00129-1, 2000.**

- P6, L166-167: I appreciate the fact that the authors performed a hyperparameter tunning instead of using default values.

That was because of the outputs were more sensitive to epoch, we only tuned this parameter. In the RM, most of the hyperparameters were tuned, and the results were improved. The hyperparameters to be optimized and their value ranges are shown in Table A2.

L340-348: **We tuned hyperparameters, which allow the learning algorithm to run until the error from the model, i.e., the loss function, has been sufficiently minimized. As there are no given values to set these numbers, the optimum values were obtained by multiple trial-and-error tests (see Table A5).**

**Table A2. The hyperparameter settings of the LSTM model**

| hyperparameter | values |
| --- | --- |
| Train portion | 65 % |
| Test  portion | 35 % |
| Epoch | 1…30 |
| Batch size | **[12, 24, 48, 72, 96, 120]** |
| Optimizer | ADAM |
| **Units        (hidden layer)** | **2…10** |
| **Dropout rate** | **0.001** |
| **Learning rate** | **0.001…0.1** |
| Loss function | MSE |

**Table A5. The optimum units, dropout, learning rate and batch size to perform the LSTM model. Here, the T and F refer to True and False.**

| models | CAMSRA | | CAMSFC | |
| --- | --- | --- | --- | --- |
| **Stations' number** | **S** | **M** | **S** | **M** |
| 1 | **10, T, 0.04, 24** | **10, T, 0.04, 24** | **2, T, 0.09, 48** | **4, F, 0.04, 48** |
| 2 | **4, F, 0.04, 48** | **4, F, 0.04, 48** | **10, T, 0.04, 24** | **10, T, 0.04, 24** |
| 3 | **10, T, 0.04, 24** | **4, F, 0.04, 48** | **2, T, 0.09, 48** | **4, F, 0.04, 48** |
| … | … | … | … | … |

- P6, L175-176: Regarding the training of the LSTM model:

Are the data shuffled prior to the training process?

To preserve the logical sequence of the data, the data was not shuffled. But since the shuffling happens on the batches axis and not on the time axis, this parameter is set to True in the RM.

Did you apply "early stopping" during training to help avoid overfitting?

Good point for the choice of epochs number. This parameter was applied in the RM.

- P9, L250-251: "That might reflect that the more predictors, the better the model would not be." Or that the M term is of less complexity and easier to be modeled?

Yes, that is right.

**L351-352: That might reflect that the M component is easier to be modeled due to less complexity.**

- P26, Table 2: How should someone interpret the fact that for station 23 only Q (specific humidity? Not included in Table A3) is important. Moreover, there are stations that the $O_3^{RA}$ (or $O_3^{FC}$ in Table A4) is not important; how should someone also interpret this?

The Q was a typo; it should be named SH. It was corrected in Table 2 and Table A4. Station 23, i.e., Rasht (known as the city of rain), is located between the mountains (Alborz) and coast (Caspian Sea). Its local environment is rainy with a humid subtropical climate. So, the humidity has a large effect on the ozone level at this site, similar to the Western Mediterranean regions. The variability of observed $O_3$ at those stations are more associated with another parameters such as NO or $NO_2$ (than $O_3^{RA}$ or $O_3^{FC}$) as shown in Fig. 4. These points were added as:

**L328-333: There are a few stations where $O_3^{RA}$ ($O_3^{FC}$) is not selected as an important variable, which is related to the small (weak) associations between $O_3^{RA}$ ($O_3^{FC}$) and $O_3^{OBS}$. For instance, SH is selected as the main factor effecting the M term at station 23, i.e., Rasht. This station is located between the mountains (Alborz) and coast (Caspian Sea), with a local environment of rainy with a humid subtropical climate. That is similar to the Western Mediterranean regions, where a lack of strong synoptic advection, combined with the orographic characteristics and the land-sea breezes, favors episodes of high ozone levels over this region (Millan et al., 1999; Velchev et al., 2011; Wentworth et. al., 2015).**

**Millán, M. M., Mantilla, E., Salvador, R., Carratalá, R., Sanz, M. J., Alonso, L., Gangioti, G., and Navazo, M.: Ozone cycles in the Western Mediterranean basin: interpretation of monitoring data in complex coastal terrain, J. Appl. Meteorol., 39, 487–508, 1999.**

**Velchev, K., Cavalli, F., Hjorth, J., Marmer, E., Vignati, E., Dentener, F., and Raes, F.: Ozone over the Western Mediterranean Sea – results from two years of shipborne measurements, Atmos. Chem. Phys., 11, 675–688, doi:10.5194/acp-11-675-2011, 2011.**

**Wentworth, G. R., Murphy, J. G., Sills, D. M. L.: Impact of lake breezes on ozone and nitrogen oxides in the greater toronto area, Atmospheric Environment, 109, 52-60, doi: 10.1016/j.atmosenv.2015.03.002, 2015.**

▪ P29, Table A3: The sea surface temperature is listed here as a meteorological variable for CAMSFC. As the SST fields are only over sea, for which coordinates are the SST data extracted for each station?

The SST was treated as other variables. We noticed that its value are fixed to 273.16 K over the land. It was removed from the list of meteorological variables in RM.

▪ A small paragraph on what might drive the differences between $O_3^{FC}$ and $O_3^{RA}$ in the discussion is needed.

The main difference between $O_3^{RA}$ and $O_3^{FC}$ was their model resolutions. We expected that the model with a finer resolution (i.e., $O_3^{FC}$) would provide better results. Our study shows more or less similar results for both models (at least for the S and M variability). Their difference might be more apparent in the bias term. Another main differences of two products, are their emissions inventories. A couple of species, i.e., $NO_x$ and CO, of the CAMS-GLOB-ANT and MACCity were assessed to show their differences. These points were added as:

L462-470: **Although both datasets share many of the same parameters in common, there are several differences that distinguish $O_3^{RA}$ from $O_3^{FC}$. $O_3^{FC}$ is produced by a model with finer horizontal and vertical resolutions. Different anthropogenic and biogenic emissions have been used in both models (see Table 1). CAMS-GLOB-ANT (used in CAMSFC) provides up-to-date emissions of air pollutants and greenhouse gases, at the spatial and temporal resolution required by the model (0.1°x0.1°). CAMSRA uses MACCity emission inventory with a resolution of 0.5°x0.5°. Figure _ shows a comparison of CAMS-GLOB-ANT and MACCity for a couple of ozone precursors, i.e., $NO_x$ and CO. Compared to CAMS-GLOB-ANT, MACCity, based on scenario, provides a higher emissions for both species. CAMS-GLOB-ANT shows more details of the emissions variability due to finer spatial resolution. In both inventories, the highest emissions area are located over Tehran.**

[Figure]

**Figure A6. The annual average of surface emissions of the (a)-(b) $NO_x$ and (c)-(d) CO in the CAMS-GLOB-ANT and MACCity emission inventories.**

- P12, L364-365: "The most relevant peroxides were found by screening several meteorological variables and chemical species." I don't understand this sentence. Please explain or rephrase.

  Sure. It was modified as:

  L580-581: **The potential predictors (inputs) were identified from chemical and meteorological variables at each station.**

**Minor comments**

- L9: Atmospheric -> Atmosphere. This is by CAMS definition.

  That was corrected.

- L11: datasets -> time series

  That was modified.

- L20: please delete ", or tropospheric ozone at ground level,"

  That was deleted.

- L30: level is -> levels are

  That was modified.

- L36: provide: provides

  That was corrected.

- L37: satellite-> satellites, and also remove "computer"

  These were modified.

- L38: technique-> techniques

That was corrected.

- L90-91: 50 chemical species and seven different aerosols. It provides outputs for several meteorological variables as well. -> 50 chemical and seven aerosol species, providing also several meteorological parameters.

  It was modified as

  L121-122: **The forecast consists of more than 50 chemical and seven different aerosols, providing also several meteorological parameters.**

- L101: . They-> and

  It was modified as

  L142: **accessible to the public and were obtained from the Iranian …**

- L127: Both reanalysis and forecast datasets were..

  It was modified as

  L169: **Both reanalysis and forecast datasets were …**

- L196: Add equation number.

  It was added.

- L224: was extracted-> were extracted

  That was corrected.

- L247 explained variability-> explained variance . Please apply this were applicable.

  A good point, that was modified as

  L349: **indicates the amount of explained variance by the LSTM model**

L350: **that more than 50 % of the $O_3$ variance is explained by the LSTM**

L426: **S in comparison to that of M is attributed to the larger variance of S (Fig. A2c)**

L433-435: **That reflects the CAMS products, which explain more of the S variance than that of the M term. In other words, most of the variance of the M term in the LSTM is explained by the lagged $O_3$ (not by the CAMS products)**

L439: **This is similar to the MSE of the LSTM, which is related to the higher variance of S than M.**

- L260: tiny->small

  Its sentences was deleted.

- L260-261: so both models show similar performance. -> with both models exhibiting similar performance.

  That was deleted.

- L307: SDS?

  A typo; that was corrected as

  L475: **by the SD procedures**

---

## Author Response (AR2)

**Authors' response to reviewer #1**

Please note that in this document, the referee's comments are in black, the authors' responses are in blue.

I thank the authors for carefully addressing my comments.

Thank you for taking the necessary time and effort to review the answers.

All is looking good, with one small exception: the sentence on line 88-89, referring to the CAMS Reanalysis: "CAMS carries several upgrades, such as improving horizontal resolutions, vertical levels, and the newer version of the satellite retrievals. CAMS uses various satellite observations, covering different time periods. " is a little confusing, and rather specifies the procedure for CAMSFC (as already described there). The whole point of the CAMS Reanalysis is that it tries to minimize any such upgrade changes, to establish a consistent time series. The system is updated only once every 8 years or so - i.e. not really relevant in this context to refer to 'upgrades'. There is no need to refer to a predecessor of CAMSRA. I'd suggest to clarify, or simply remove the sentence completely. If this is corrected then the manuscript can be published as such.

To be on the safe side, the sentence was removed.

**Authors' response to reviewer #2**

Please note that in this document, the referee's comments are in black, the authors' responses are in blue.

Overall, the authors addressed my comments. I only have a few minor comments for the revised manuscript and suggest publication in GMD.

We thank the reviewer for taking the necessary time an effort to review the responses.

P1, l15-16: Results show the benefit of applying the LSTM method instead of using the original CAMS products for providing O3 over Iran.
That was modified.

P3, l74: "These datasets specific to tropospheric ozone analysis". What do you mean here? Maybe "These datasets focusing on surface ozone"?
We mean that the focus is on the surface ozone. That was modified.

P3, l80: IFS (CB05)
That was modified.

P10-11, l312-314: "The S variability results from the effect of daytime photochemical production, downward transport of O3 rich from upper levels, combined with O3 loss by depositions (in the surface layer)". "Results from" is somehow strong here, maybe "is associated with"..
That was changed.

P13, L400: upper troposphere in larger time scales (Zanis et al., 2014) Please also include here the study by Akritidis et al. (2016) (it was included in the initial manuscript), as it highlights the role of stratospheric ozone transport in tropospheric ozone over the examined region also.
References
Akritidis, D., Pozzer, A., Zanis, P., Tyrlis, E., Škerlak, B., Sprenger, M., and Lelieveld, J.: On the role of tropopause folds in summertime tropospheric ozone over the eastern Mediterranean and the Middle East, Atmos. Chem. Phys., 16, 14025–14039, https://doi.org/10.5194/acp-16-14025-2016, 2016
That was added.

P5, l145: "stratospheric-tropospheric exchange" -> stratosphere-troposphere exchange
That was corrected.

Conclusions: I suggest using bullet numbering or simple bullets instead of numbers in the form (1) (2) ...

The numbers were replaced with bullet points.

P15, l457-459: I suggest to modify as follows: "In addition, uncertainties in emission inventories might affect this error. Including a proxy of stratospheric ozone contribution to surface ozone (stratospheric ozone tracer) may be beneficial in explaining sort term ozone variability, thus reducing the error (a recommendation for future work)."

That was applied.

P15, l460: belongs to -> appears for

That was changed.

Table 1 caption: remove "some"

That was removed.